# Incomplete activation of *Alyref* and *Gabpb1* leads to preimplantation arrest in cloned mouse embryos

Shunya Ihashi[1], Mizuto Hamanaka[1,*], Masaya Kaji[1,*], Ryunosuke Mori[1,*], Shuntaro Nishizaki[1,*], Miki Mori[1], Yuma Imasato[1], Kimiko Inoue[2,3], Shogo Matoba[2,4], Narumi Ogonuki[2], Atsushi Takasu[1], Misaki Nakamura[1], Kazuya Matsumoto[1], Masayuki Anzai[5], Atsuo Ogura[2,3], Masahito Ikawa[6], Kei Miyamoto[1]

Differentiated cell nuclei can be reprogrammed after nuclear transfer (NT) to oocytes and the produced NT embryos can give rise to cloned animals. However, development of NT embryos is often hampered by recurrent reprogramming failures, including the incomplete activation of developmental genes, yet specific genes responsible for the arrest of NT embryos are not well understood. Here, we searched for developmentally important genes among the reprogramming-resistant H3K9me3-repressed genes and identified *Alyref* and *Gabpb1* by siRNA screening. Gene knockout of *Alyref* and *Gabpb1* by the CRISPR/Cas9 system resulted in early developmental arrest in mice. *Alyref* was needed for the proper formation of inner cell mass by regulating *Nanog*, whereas *Gabpb1* deficiency led to apoptosis. The supplement of *Alyref* and *Gabpb1* mRNA supported efficient preimplantation development of cloned embryos. *Alyref* and *Gabpb1* were silenced in NT embryos partially because of the repressed expression of *Klf16* by H3K9me3. Thus, our study shows that the H3K9me3-repressed genes contain developmentally required genes, and the incomplete activation of such genes results in preimplantation arrest of cloned embryos.

## Introduction

After fertilization, germ cells acquire totipotency, the ability to give rise to all cells that build the conceptus. On the other hand, a somatic cell nucleus can be reprogrammed to be totipotent after transplanting it into an enucleated oocyte. This technique is called somatic cell nuclear transfer (SCNT). Using SCNT, the successful cloning of an animal was first reported in *Xenopus laevis* (Gurdon, 1962), followed by various mammalian species such as sheep, mice, and cattle (Wilmut et al, 1997; Cibelli et al, 1998; Wakayama et al,

1998). However, the efficiency of obtaining cloned animals by conventional SCNT methods is low, and a myriad of defects have been reported during both pre and postimplantation development (Ogura et al, 2013; Loi et al, 2016). The low efficiency of SCNT is one of the main factors that impede the prevalence of this technology, even though SCNT potentially offers an invaluable opportunity for basic biology, regenerative medicine, and the propagation of endangered animals.

The molecular basis for the low developmental potential of SCNT embryos has been investigated. It is now generally recognized that the abnormal state of histone modifications in SCNT embryos is a major cause of aberrant gene expression and low developmental ability. Facilitation of histone acetylation by the addition of trichostatin A (TSA), a small molecule compound that inhibits histone deacetylases, into the embryo culture medium has been widely applied for enhancing the development of SCNT embryos (Kishigami et al, 2006; Rybouchkin et al, 2006). It has also been reported that histone H3 lysine 9 trimethylation (H3K9me3) accumulated in somatic genomes persists after SCNT and prevents zygotic genome activation (ZGA) of a subset of genes (Matoba et al, 2014). Inhibitory effects of H3K9me3 on nuclear reprogramming in SCNT embryos have been confirmed in several species (Chung et al, 2015; Liu et al, 2018). Furthermore, the persisting H3K9me3 mark prevents nuclear reprogramming in induced pluripotent stem cells (Soufi et al, 2012), demonstrating H3K9me3 as a universal barrier to reprogramming. In other words, H3K9me3 serves as a solid mechanism to repress lineage-inappropriate genes in somatic cell genomes (Becker et al, 2016), and it would be important to experimentally test if genes required for accomplishing reprogramming are indeed included among H3K9me3-repressed genes.

In our previous study, we found that when mouse SCNT embryos are treated with TSA and then cultured in vitamin C (VC)-supplemented medium, 83.6% of SCNT embryos develop to the blastocyst stage and 15.2% of SCNT embryos develop into offspring (Miyamoto et al, 2017),

[1]Laboratory of Molecular Developmental Biology, Faculty of Biology-Oriented Science and Technology, Kindai University, Wakayama, Japan    [2]Bioresource Engineering Division, RIKEN Bioresource Research Center, Tsukuba, Japan    [3]Graduate School of Life and Environmental Sciences, University of Tsukuba, Tsukuba, Japan    [4]Cooperative Division of Veterinary Sciences, Tokyo University of Agriculture and Technology, Fuchu, Japan    [5]Institute of Advanced Technology, Kindai University, Wakayama, Japan    [6]Research Institute for Microbial Diseases, Osaka University, Suita, Japan

Correspondence: kmiyamo@waka.kindai.ac.jp
*Mizuto Hamanaka, Masaya Kaji, Ryunosuke Mori, and Shuntaro Nishizaki contributed equally to this work

representing one of the most efficient development of cloned embryos (Ogura, 2020). This treatment with VC lowers the level of H3K9me3 and significantly alters transcriptomes in SCNT embryos (Miyamoto et al, 2017). It is therefore plausible that reprogramming-resistant genes regulated by H3K9me3 are up-regulated by the TSA and VC treatments, which might have eventually resulted in the enhanced development of SCNT embryos. In this article, we found that among previously identified reprogramming-resistant genes (Matoba et al, 2014), 16 genes were up-regulated in SCNT embryos under TSA- and VC-treated conditions. We further performed siRNA screening to identify developmentally important genes, and *Alyref* and *Gabpb1* were identified. Alyref is an mRNA-binding adaptor protein involved in nuclear export of mRNA (Zhou et al, 2000; Rodrigues et al, 2001). Alyref interacts with lws1 to mediate mRNA transport (Yoh et al, 2007), and it has been reported that the knockdown of lws1 reduces mRNA transport and arrests embryonic development at 8–16 cell stages (Oqani et al, 2019). *Gabpb1* encodes Gabp-β, one of the subunits of GA-binding protein (GABP) that acts as a transcriptional activator (Rosmarin et al, 2004). GABP is known to activate the *Yap* gene, which is necessary for preimplantation development (Wu & Guan, 2021), and the knockdown of GABP has been reported to decrease YAP and inhibit progression to the G1/S stage, eventually leading to increased cell death (Wu et al, 2013). However, roles of *Alyref* and *Gabpb1* in preimplantation development remain unclear. Here, we showed that knockout of *Alyref* and *Gabpb1* caused early embryonic arrest in mouse fertilized embryos. Furthermore, the expression of Alyref and Gabpb1 was repressed in SCNT embryos at the protein level, and the rescued expression of Alyref and Gabpb1 by mRNA injection enhanced preimplantation development of SCNT embryos. Therefore, *Alyref* and *Gabpb1*, genes necessary for preimplantation development, are normally repressed in SCNT embryos, and the incomplete activation of *Alyref* and *Gabpb1*, at least partially, explains the low developmental capacity of SCNT embryos.

# Results

## Identification of developmentally important genes that are down-regulated in SCNT embryos

We have previously shown that development of SCNT embryos is greatly enhanced by the combinational treatment of TSA and VC (Miyamoto et al, 2017). We hypothesized that these epigenetic modifiers allowed activation of key genes for embryonic development, which were otherwise repressed in SCNT embryos. We therefore compared down-regulated genes in control SCNT embryos at the late two-cell stage (control SCNT versus SCNT with TSA and VC) (Miyamoto et al, 2017) to those found between SCNT embryos and in vitro fertilized (IVF) embryos (a group of 301 genes that failed to be activated in SCNT embryos) (Matoba et al, 2014). Among the down-regulated genes, 16 genes were shared (Fig 1A and B). These genes are normally repressed in SCNT embryos, but are up-regulated when the development of cloned embryos is greatly enhanced by the TSA + VC treatment. We then searched for developmentally important genes among the list by knocking down

their expression in IVF embryos with siRNA injection and specific siRNA sets for 15 genes were designed. The siRNA screening identified that knockdown of *Alyref* and *Gabpb1* significantly impaired the development of IVF embryos to the blastocyst stage (Fig 1C). Successful knockdown of both genes was shown by qRT-PCR (Fig S1A). We also observed a significant decrease in preimplantation development after injecting the mixture of siRNAs that target genes, which have been shown to function as transcriptional activators including *Gabpb1*, *Gtf2f2*, and *Taf9* (Rosmarin et al, 2004) (Fig 1C; activators). Inhibition of *Alyref* or *Gabpb1* by siRNA injection resulted in developmental arrest at the morula stage (Figs 1D and S1B); especially, almost all embryos did not develop to the blastocyst stage in *Alyref* siRNA-injected embryos (Fig S1B). These results suggest that *Alyref* or *Gabpb1* genes, repressed in SCNT embryos, are important for the progression to the blastocyst stage.

## Abnormal expression of Alyref and Gabpb1 in SCNT embryos

We next examined if the abnormal expression of Alyref or Gabpb1 is indeed observed in SCNT embryos. Firstly, the protein localization of Alyref and Gabpb1 was examined during the preimplantation development. Alyref and Gabpb1 were localized in pronuclei of mouse IVF zygotes (Fig 2A and B), suggesting maternal accumulation of Alyref and Gabpb1 protein. Nuclear localization was seen throughout the preimplantation development. To note, dot-like localization of Alyref was observed in the nuclei of IVF embryos from the two-cell stage onwards (Figs 2A and S1C). Gabpb1 protein was accumulated around nucleoli from the four-cell stage onwards (Fig 2B). Secondly, the transcriptional expression of *Alyref* or *Gabpb1* in SCNT embryos was examined using RNA-seq dataset in Matoba et al (2014) and Miyamoto et al (2017). Both *Alyref* and *Gabpb1* were repressed in SCNT embryos compared with IVF embryos at the two-cell stage and this repression was rescued by the overexpression of Kdm4d (Fig S1D), which removes excess H3K9me3 observed in SCNT embryos (Matoba et al, 2014). Up-regulation of *Alyref* and *Gabpb1* was also observed in SCNT embryos at the late two-cell stage treated with TSA and VC (Fig S1D), which also lowers the level of H3K9me3 (Miyamoto et al, 2017). Repression of *Alyref* and *Gabpb1* in SCNT embryos at the late two-cell stage, as compared with TSA-VC–treated SCNT embryos, was confirmed by qRT-PCR analyses (Fig 2C). These results suggest that *Alyref* and *Gabpb1* are repressed at the transcript level in SCNT embryos at the time of ZGA. Lastly, the protein expression of Alyref and Gabpb1 in SCNT embryos was examined at 28, 48, and 72 h post-activation (hpa). Both Alyref and Gabpb1 were significantly down-regulated in SCNT embryos at all stages tested when compared with IVF embryos (Fig 2D and E). These results suggest that abnormal down-regulation of Alyref and Gabpb1 is observed in SCNT embryos both at the transcript and protein levels.

## *Alyref* gene is essential for preimplantation development

To clarify the requirement of *Alyref* for embryonic development, we generated knockout mice by the CRISPR/Cas9 gene-editing method. The coding region of *Alyref* gene was removed by introducing gRNAs and Cas9 protein to zygotes as reported before (Mashiko et al, 2013)

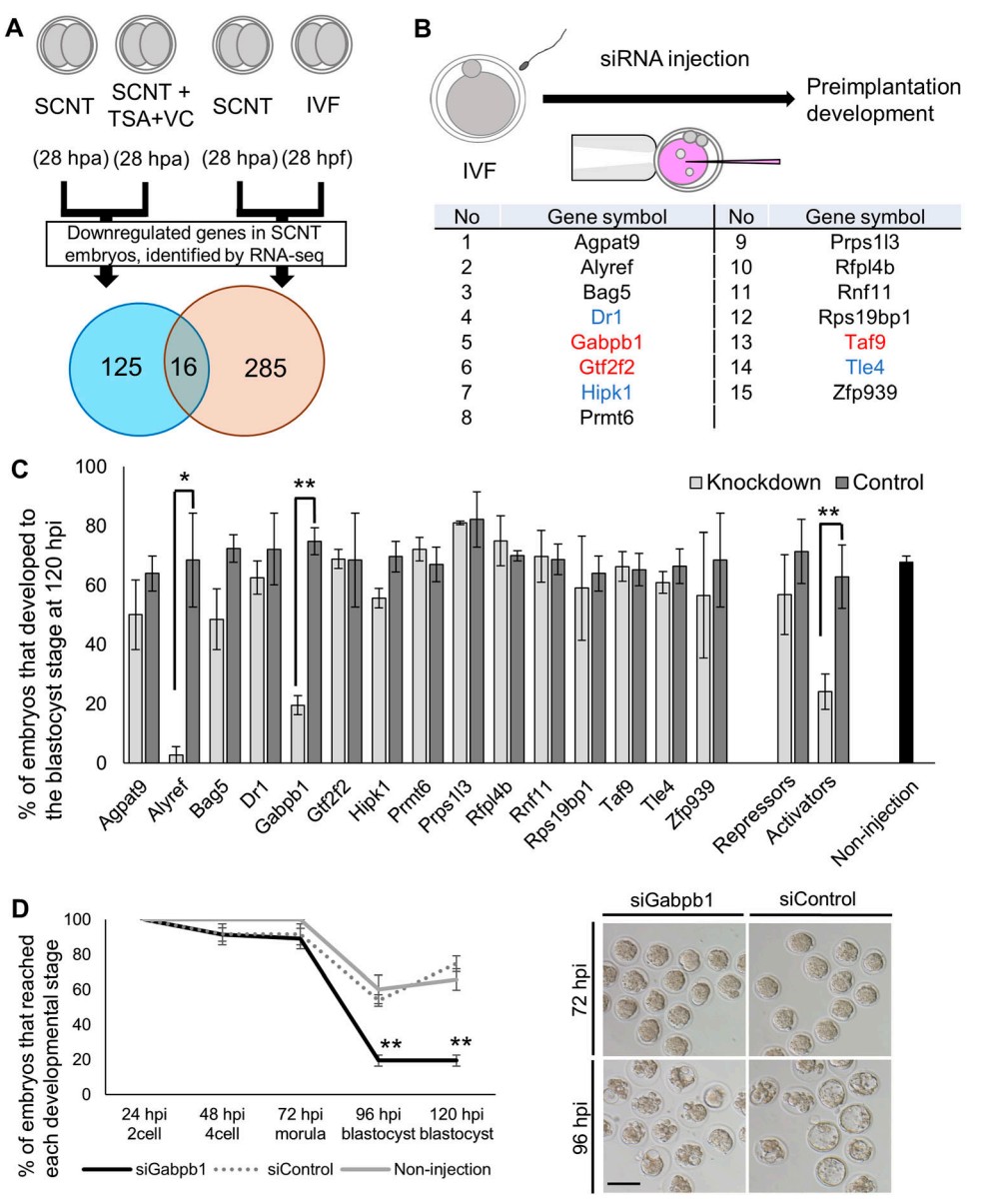

**Figure 1. Screening for candidate genes is key for development of mouse somatic cell nuclear transfer (SCNT) embryos.**
**(A)** A strategy for identifying genes responsible for reprogramming in SCNT embryos and those for development in normal in vitro fertilized embryos. hpa, hours post-activation; hpf, hours post-fertilization. **(B)** An experimental scheme for siRNA screening to find genes important for the development of fertilized embryos and the list of candidate genes. Genes that function as transcriptional activators are shown in red, whereas those that function as transcriptional repressors are in blue. **(C)** Development of embryos injected with the listed siRNAs. Control represents embryos injected with control siRNA. "Repressors" include genes marked with blue in Fig 1B, whereas "Activators" include those with red. hpi, hours post-insemination. $N$ = siAgpat9: 3, siAlyref: 3, siBag5: 8, siDr1: 3, siGabpb1: 4, siGtf2f2: 3, siHipk1: 4, siPrmt6: 3, siPrps1l3: 4, siRfpl4b: 3, siRnf11: 4, siRps19bp1: 4, siTaf9: 4, siTle4: 4, siZfp939: 3, siRepressors: 3, siActivators: 3. Non-injection: 48. $n$ = siAgpat9: 36 and 31, siAlyref: 32 and 25, siBag5: 103 and 84, siDr1: 31 and 26, siGabpb1: 55 and 48, siGtf2f2: 34 and 25, siHipk1: 36 and 28, siPrmt6: 35 and 34, siPrps1l3: 37 and 31, siRfpl4b: 45 and 51, siRnf11: 41 and 42, siRps19bp1: 37 and 31, siTaf9: 36 and 33, siTle4: 38 and 40, siZfp939: 35 and 25, siRepressors: 33 and 37, siActivators: 36 and 38 (indicated as siRNAs against target genes and control siRNAs, respectively), Non-injection: 346. **(D)** Preimplantation development of embryos injected with siRNA against *Gabpb1*, control siRNA, and non-injected embryos. Representative images of the injected embryos are shown in the right panel. $N$ = 4. $n$ = siGabpb1: 55, siControl: 48, and non-injection: 21. Scale bar = 100 $\mu$m. Data information: bars represent mean ± SEM. $N$ number refers to independent injection experiments. $n$ number refers to the number of embryos used for each treatment. **(C, D)** *P < 0.05, **P < 0.01, determined by two-sided F- and t tests for (C) and chi-square test for (D).

---

(Fig 3A). The resulting heterozygous mutant mice (Alyref$^{+/-}$) were identified by PCR-based genotyping (Fig S2A). When Alyref$^{+/-}$ mice were mated with WT mice, viable offspring was obtained (Fig 3B). However, the number of offspring was significantly reduced (Fig S2B) and no knockout mice (Alyref$^{-/-}$) were obtained after crossing Alyref $^{+/-}$ × Alyref$^{+/-}$ (Fig 3B), suggesting the embryonic lethal phenotype of *Alyref* knockout mice. We then performed IVF using sperm and oocytes obtained from Alyref $^{+/-}$ mice to reveal when development is arrested. The significant decrease in percentages of embryos that reach the blastocyst stage was observed (Fig 3C and D). Immunofluorescent analyses revealed that all blastocyst embryos obtained by Alyref $^{+/-}$ × Alyref $^{+/-}$ showed Alyref signals, whereas some of the morula embryos lacked Alyref expression (Fig S2C). These results demonstrate that *Alyref* is required for mouse

preimplantation development and knockout of *Alyref* causes developmental arrest at the morula stage.

## *Gabpb1* gene is essential for embryonic development

To test the requirement of *Gabpb1* for embryonic development, we generated knockout mice by the CRISPR/Cas9 method. gRNA that targets exon 3 of *Gabpb1* was designed and introduced to zygotes with Cas9 protein (Fig 4A). The resulting heterozygous mutant mice (Gabpb1$^{+/-}$) were identified by DNA sequencing (Fig S2D) and the frame shift mutation was confirmed in the knockout allele (Fig 4B), which resulted in a truncated protein. When Gabpb1$^{+/-}$ mice were mated with WT mice, viable offspring was obtained (Fig 4C and D). However, when Gabpb1$^{+/-}$ mice were mated with Gabpb1$^{+/-}$, the

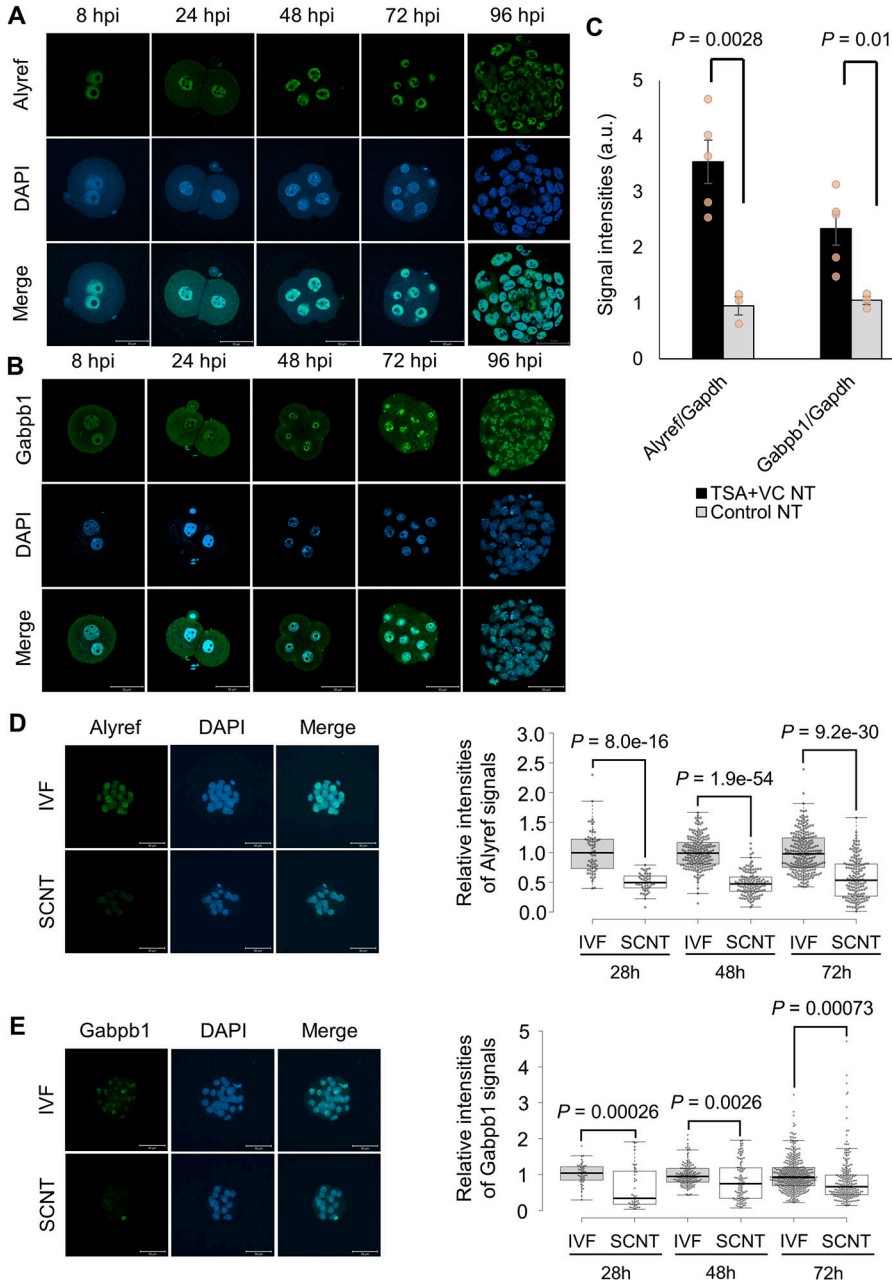

**Figure 2. Expression of Alyref and Gabpb1 in fertilized and cloned embryos.**
**(A)** Immunostaining of Alyref in fertilized embryos during preimplantation development. DNA was stained with DAPI. Merge represents the merged photos. *N* = 8 hpi: 4, 24 hpi: 5, 48 hpi: 5, 72 hpi: 6, 96 hpi: 5. *n* = 8 hpi: 35, 24 hpi: 34, 48 hpi: 32, 72 hpi: 40, 96 hpi: 36. Scale bars = 50 μm. **(B)** Immunostaining of Gabpb1 in fertilized embryos during preimplantation development. *N* = 8 hpi: 6, 24 hpi: 7, 48 hpi: 5, 72 hpi: 6, 96 hpi: 4. *n* = 8 hpi: 62, 24 hpi: 38, 48 hpi: 54, 72 hpi: 64, 96 hpi: 46. Scale bars = 50 μm. **(C)** qRT-PCR analyses of two-cell somatic cell nuclear transfer (SCNT) embryos at 28 hpa treated with (black) or without TSA + VC (grey). Relative transcript levels of *Alyref* or *Gabpb1* to *Gapdh* were measured. TSA + VC NT: 5 and control NT: three biological replicates. **(D)** Immunostaining of Alyref at the morula stage (left panel, 72 hpi) and the box plot indicates signal intensities of Alyref in in vitro fertilized (IVF) and SCNT embryos. SCNT and IVF embryos were collected at 28, 48, 72 hpa and 28, 48, 72 hpi, respectively. *N* = 4; *n* = SCNT: 24 (28 hpa), 31 (48 hpa), 15 (72 hpa); and IVF: 30 (28 hpi), 25 (48 hpi), 23 (72 hpi). Each plot represents the nuclear signal intensity. Scale bars = 50 μm. **(E)** Immunostaining of Gabpb1 at the morula stage (left panel, 72 hpi) and the box plot indicates signal intensities of Gabpb1 in IVF and SCNT embryos. SCNT and IVF embryos were collected at 28, 48, 72 hpa and 28, 48, 72 hpi, respectively. *N* = 4 (28, 48 hpa, 28, and 48 hpi samples) and 5 (72 hpa and hpi samples). *n* = SCNT: 23 (28 hpa), 25 (48 hpa), 23 (72 hpa) and IVF: 24 (28 hpi), 21 (48 hpi), 40 (72 hpi). Each plot represents the nuclear signal intensity. Scale bars = 50 μm. Data information: bars represent mean ± SEM. *N* number refers to independent experiments. *n* number refers to the numbers of embryos used for each treatment. Statistical significance (*P*-values) was determined by two-sided F- and t tests.

number of offspring was significantly reduced (Fig 4D) and no knockout mice (Gabpb1$^{-/-}$) were obtained (Fig 4C). We then carried out IVF using sperm and oocytes obtained from Gabpb1$^{+/-}$ mice. The significant decrease in percentages of embryos that reach the morula and blastocyst stages was observed (Fig 4E). Nevertheless, a certain number of embryos reached blastocysts after crossing Gabpb1$^{+/-}$ × Gabpb1$^{+/-}$, and we have therefore performed genotyping of the generated embryos. Many of Gabpb1$^{-/-}$ embryos (75%) were arrested at the morula stage or degenerated, whereas 61% of Gabpb1$^{+/-}$ embryos developed to the blastocyst state (Fig S2E). The lack of Gabpb1$^{-/-}$ expression in the arrested morula embryos was also confirmed by immunofluorescent analyses (Fig 4F). These

results demonstrate that *Gabpb1* is required for mouse embryonic development.

## Developmentally important pathways are disturbed after knocking out *Alyref* or *Gabpb1*

To gain mechanistic insight into the role of *Alyref* and *Gabpb1*, we performed RNA sequencing (RNA-seq) analysis. Based on the IVF experiments using sperm and oocytes of heterozygous mutant mice, most of knockout embryos were arrested at the morula stage at 96 h post-insemination (hpi) (Figs 3 and 4). Therefore, embryos remained as morulae at 96 hpi were collected and subjected to

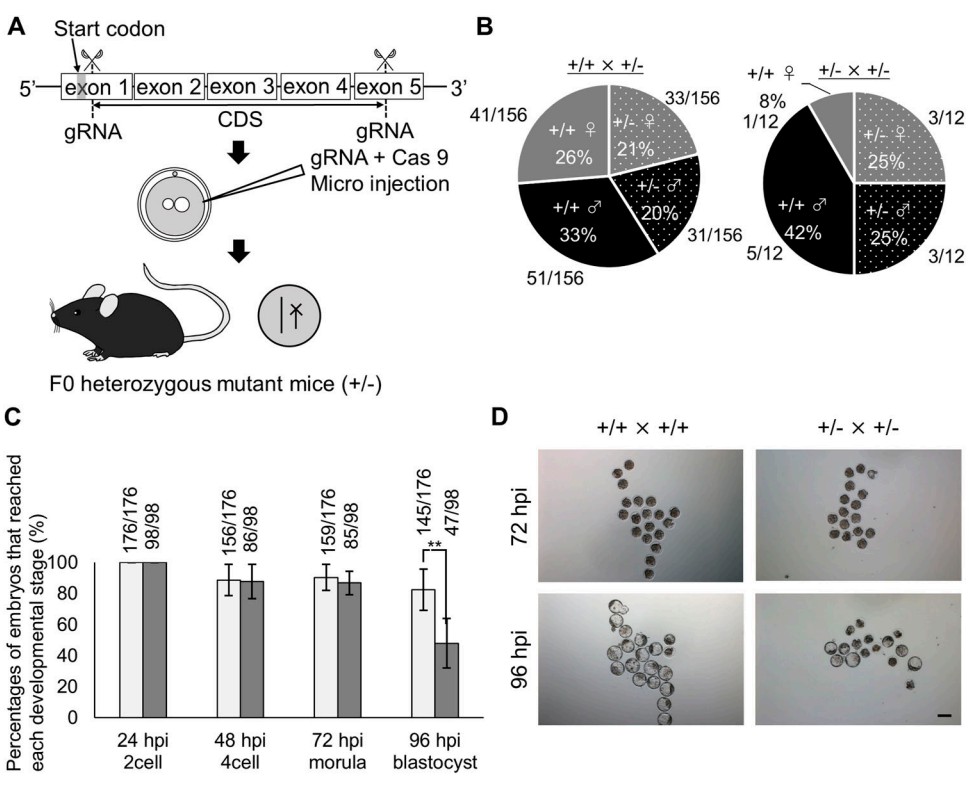

**Figure 3. *Alyref* is required for mouse preimplantation development.**
**(A)** A scheme for CRISPR/Cas9-mediated knockout of *Alyref* gene. Two gRNAs were introduced with Cas9 protein to mouse zygotes to induce the large deletion of the coding region of *Alyref*. **(B)** Pie charts represent percentages of genotypes and sex of the offspring after each mating (Alyref[+/+] × Alyref[+/−] [left] and Alyref[+/−] × Alyref[+/−] [right]). The numbers of offspring are also shown. **(C)** Preimplantation development of WT embryos (Alyref[+/+] × Alyref[+/+]) and embryos generated by mating heterozygous mutant mice (Alyref[+/−] × Alyref[+/−]). The numbers of embryos that reached each developmental stage are indicated above the bars. $N$ = 6. **(D)** Representative images of embryos. Control Alyref[+/+] × Alyref[+/+] embryos are shown in left panels, whereas Alyref[+/−] × Alyref[+/−] embryos are shown in the right. Scale bar = 100 μm. Data information: bars represent mean ± SEM. $N$ number refers to independent experiments. \*\*$P$ < 0.01, determined by chi-square test.

single-embryo RNA-seq together with control blastocyst embryos (Fig 5A). The large deletion of *Alyref* was confirmed in two morula embryos (Fig 5B; KOA-M1 and KOA-M2) because no sequence reads were detected at the deleted region. Similarly, the frameshift mutation of *Gabpb1* was also confirmed in three morula embryos (Fig 5C; KOG-M1-M3). Off-target deletion by CRISPR/Cas9 was not observed in *Alyref2* and *Gabpb2*, family genes of *Alyref* and *Gabpb1*, respectively (Fig S3A and B). Principal component analysis and hierarchical clustering analysis indicated that knockout morulae showed distinct transcriptomes from WT morulae and were well separated from those of blastocyst embryos (Figs 5D and E and S3C and D). These results suggest that knockout of *Alyref* and *Gabpb1* causes abnormal gene expression in morulae, which may prevent the progression of knockout embryos to the blastocyst stage.

We next examined molecular pathways that were disturbed by the depletion of *Alyref*. Differentially expressed genes (DEGs) were identified between knockout and WT (and/or heterozygous) morulae (KOA-M1-2 versus A-M1-3) and 705 genes were misregulated ($P$ < 0.05, twofold changes). Ingenuity pathway analysis (IPA) using this DEG list showed that genes related to mammalian pluripotency were misregulated (Fig 6A and Appendix Table S1). We then searched for upstream regulators responsible for the abnormal gene expression in Alyref[−/−] morulae using IPA. Interestingly, *Pou5f1*, a master regulator for mammalian pluripotency, was identified (Fig 6B). We examined Pou5f1-positive cells in *Alyref* KO embryos. Expansion of strongly Pou5f1-positive cells and the proper formation of inner cell mass (ICM) were not observed in Alyref-negative embryos (Fig 6C). However, the abolishment of Oct4 signals

was not observed in *Alyref* KO embryos (Fig 6C, bottom panels). These results suggest that *Alyref* may not be a direct upstream factor to regulate *Pou5f1* expression in early embryos, but it is still possible that *Alyref* might play a role in ICM formation by cooperating with *Pou5f1* and/or other factors. To test this idea, we next focused on *Alyref*'s function to regulate *Nanog* expression as *Nanog*-related pathways were disturbed in *Alyref* KO embryos (Fig 6A). Previous studies demonstrated that Alyref is a component of the transcription/export complex, and Thoc2 and Thoc5, other components of the transcription/export complex, are required for the self-renewal of embryonic stem (ES) cells (Wang et al, 2013). Interestingly, Thoc2 and Thoc5 regulate export and expression of *Nanog* mRNA, but not *Pou5f1* in ES cells (Wang et al, 2013). We therefore tested if *Alyref* regulates the expression of *Nanog* for ICM formation in blastocyst embryos. Expression of Nanog protein was significantly reduced in *Alyref* knockdown and knockout embryos at 72 hpi (Fig 6D). Furthermore, Alyref inhibition by siRNA injection resulted in down-regulation of *Nanog* transcripts (Fig 6E). Interestingly, expression of *Nanog* intron transcripts in *Alyref* knockdown embryos was not significantly reduced, compared with that in control embryos (Fig 6E), suggesting that *Nanog* RNA processing for its translation, but not *Nanog* gene activation, may be defective after *Alyref* knockdown. Because *Gapdh* expression was also reduced in *Alyref* knockdown embryos, developmental progression of *Alyref* knockdown embryos might be delayed, which affected global transcript levels. Indeed, cell numbers were significantly reduced after *Alyref* knockdown (siAlyref embryos at 72 hpi: 11 ± 0.83 cells versus siControl embryos at 72 hpi: 18 ± 1.6 cells, $P$ < 0.01, F- and

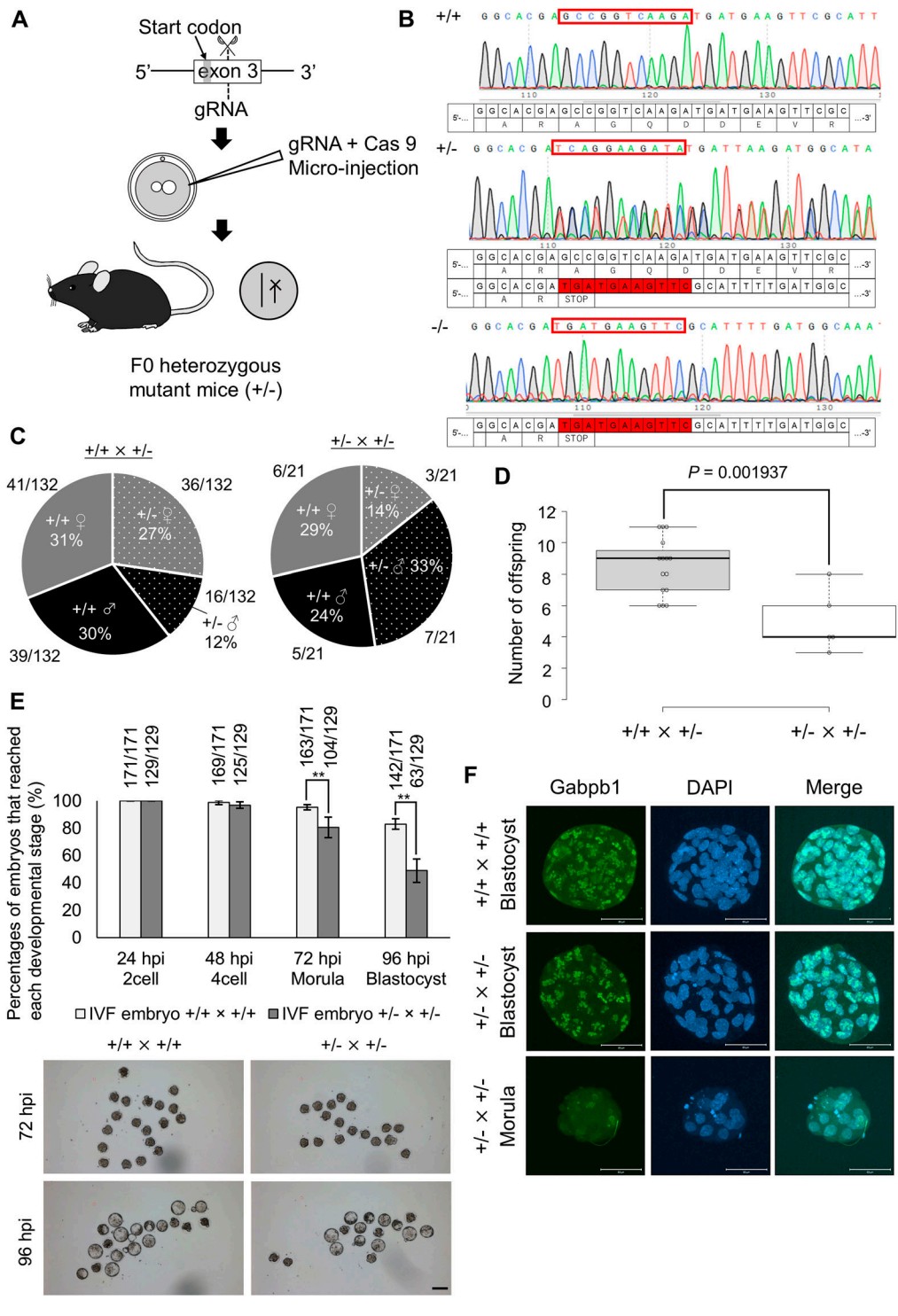

**Figure 4. *Gabpb1* is required for mouse embryonic development.**
**(A)** A scheme for CRISPR/Cas9-mediated knockout of *Gabpb1* gene. gRNA that targets exon 3 of *Gabpb1* was introduced with Cas9 protein to mouse zygotes to induce the frame-shift mutation of *Gabpb1*. **(B)** DNA-sequencing analyses show the frame-shift mutation in the knockout allele of *Gabpb1*. The deleted genomic region is highlighted in red color. **(C)** Pie charts represent percentages of genotypes and sex of the offspring after each mating (Gabpb1$^{+/+}$ × Gabpb1$^{+/-}$ [left] and Gabpb1$^{+/-}$ × Gabpb1$^{+/-}$ [right]). The numbers of offspring are also shown. **(D)** The number of offspring is shown after each mating. **(E)** Preimplantation development of WT in vitro fertilized (IVF) embryos (Gabpb1$^{+/+}$ × Gabpb1$^{+/+}$) and IVF embryos generated by using heterozygous mutant mice (Gabpb1$^{+/-}$ × Gabpb1$^{+/-}$). The numbers of embryos that reached each developmental stage are indicated above the bars. Representative images of IVF embryos are shown at the bottom (left panels: control Gabpb1$^{+/+}$ × Gabpb1$^{+/+}$ embryos, right panels: Gabpb1$^{+/-}$ × Gabpb1$^{+/-}$ embryos). *N* = 6. Scale bar = 100 μm. **(F)** Immunostaining of Gabpb1 in IVF embryos derived from different combinations of crossing. Embryos were collected at 96 hpi. DNA was stained with DAPI. *N* = 3. *n* = (Gabpb1$^{+/+}$ × Gabpb1$^{+/+}$): 102, and (Gabpb1$^{+/-}$ × Gabpb1$^{+/-}$): 82. Scale bars = 50 μm. Data information: bars represent mean ± SEM. *N* number refers to independent experiments. *n* number refers to the numbers of embryos used for each treatment. **(D, E)** \*\**P* < 0.01, determined by two-sided F- and *t* test for (D) and chi-square test for (E).

*t* test). Finally, Alyref overexpression in one of the two-cell blastomeres increased Nanog-positive cells at the blastocyst stage at 96 hpi (Fig 6F). Thus, Alyref ensures Nanog expression and is required for the developmental progression to blastocyst embryos.

To gain mechanistic insight into the developmental arrest caused by *Gabpb1* depletion, we performed hierarchical clustering of genes based on their expression levels using *Gabpb1* knockout and WT embryos and 20 clusters were generated (Fig S4A). Genes that belong to cluster 1 showed very low expression in Gabpb1$^{-/-}$ morulae, but were expressed in other samples with the strongest expression in WT blastocysts (Fig S4A). Cluster 1 was enriched with genes related to blastocyst formation, and thus, knockout of

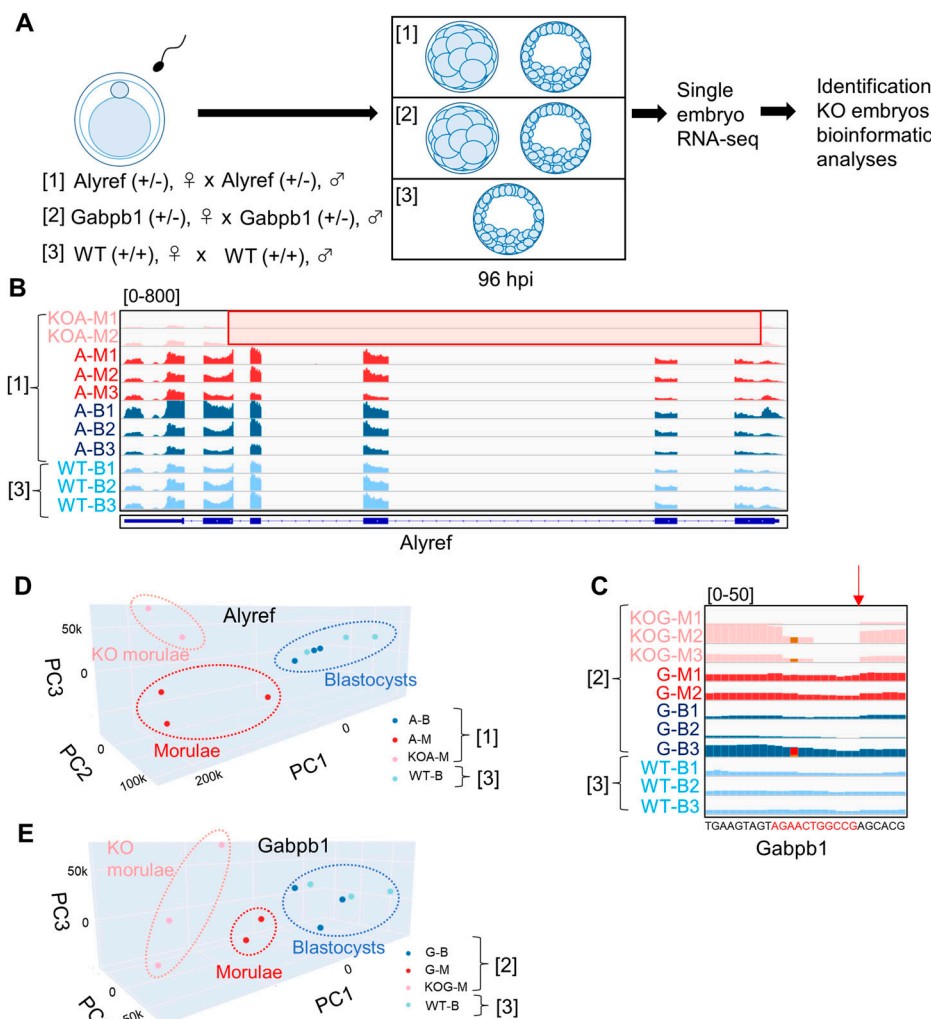

**Figure 5. Abnormal transcriptomic patterns in *Alyref*⁻/⁻ or *Gabpb1*⁻/⁻ embryos.**
**(A)** Experimental design to examine transcriptomes of *Alyref*⁻/⁻ or *Gabpb1*⁻/⁻ embryos. Three different combinations of in vitro fertilized experiments were performed (1–3), and morula or blastocyst embryo was collected from each in vitro fertilized experiment for RNA-seq analyses. **(B, C)** Track images of RNA-seq reads at *Alyref* and *Gabpb1* gene loci. **(B)** shows sequence reads of morulae and blastocysts collected after fertilization between *Alyref* heterozygous mutant mice or control WT mice as shown in (A) ([1] and [3]), whereas (C) represents those collected after fertilization between *Gabpb1* heterozygous mutant mice or control WT mice as shown in (A) ([2] and [3]). *Alyref*⁻/⁻ embryos (KOA-M1, 2) were recognized by the lack of sequence reads at the deleted region (red highlighted box). Non-knockout morula and blastocyst embryos obtained from fertilization between *Alyref* heterozygous mutant mice are shown as A-M1-3 and A-B1-3, respectively, whereas control WT blastocyst embryos are named as WT-B1-3. Similarly, *Gabpb1*⁻/⁻ embryos (KOG-M1-3) were identified because of the lack of sequence reads from the deleted locus (red arrow). Non-knockout morula and blastocyst embryos obtained from fertilization between *Gabpb1* heterozygous mutant mice are shown as G-M1, 2, and G-B1-3, respectively. **(D, E)** Principal component analysis of the global gene expression profile among different samples after fertilization between *Alyref* heterozygous mutant mice (D) and that between *Gabpb1* heterozygous mutant mice (E). WT blastocyst embryos are used as control (WT-B). PC1–3, principal components 1–3.

*Gapbp1* leads to the incomplete activation of such developmentally important genes. Next, DEGs were identified between *Gabpb1* knockout and WT morulae (KOG-M1-3 versus G-M1-2) and 1,321 genes were misregulated (*P* < 0.05, twofold changes). IPA revealed that anti-oxidation and metabolic pathways were abnormally regulated in Gabpb1⁻/⁻ embryos (Fig S4B, highlighted in green). Candidate upstream regulators related to this misexpression were identified (Fig S4C). Moreover, genes related to cell viability and apoptotic pathways showed abnormal expression (Fig 6G). We therefore examined the appearance of apoptotic cells after *Gabpb1* knockout. At 96 hpi, 39% ± 2.6% of cells underwent apoptosis when *Gabpb1* siRNA was injected (Fig 6H). Many apoptotic cells were also found in arrested morulae embryos after crossing Gabpb1⁺/⁻ × Gabpb1⁺/⁻ (Fig S4D). Furthermore, abnormal mitochondrial membrane potential was detected in *Gabpb1* knockdown embryos at 96 hpi (Fig 6I). These results indicate that suppression of *Gabpb1* leads to abnormal gene expression related to anti-oxidation, metabolic pathways, and blastocyst formation; disturbed mitochondrial functions; and eventually the induction of apoptosis, resulting in the failure to form proper blastocyst embryos. In summary, the roles of *Alyref* and *Gabpb1* in mouse preimplantation development are different, but both genes are required for the establishment of proper gene expression programs for blastocyst formation.

## The relationship between H3K9me3-mediated gene repression and expression of *Alyref* and *Gabpb1* expression

The above data have indicated that *Alyref* and *Gabpb1* are required for preimplantation development, but repressed in SCNT embryos. Expression of these genes is repressed by H3K9me3-related mechanisms because overexpression of Kdm4d and treatment with VC, both of which can lower H3K9me3 levels in SCNT embryos (Matoba et al, 2014; Miyamoto et al, 2017), can up-regulate *Alyref* and *Gabpb1* (Figs 2C and S1D). We therefore asked if the expression of *Alyref* and *Gabpb1* is directly or indirectly regulated by H3K9me3. We reanalyzed published ChIP-seq dataset in which H3K9me3 enrichment was examined in fertilized embryos, SCNT embryos at the two-cell stage, and donor cumulus cells (Liu et al, 2016). H3K9me3 was not much accumulated around *Alyref* and *Gabpb1* in donor

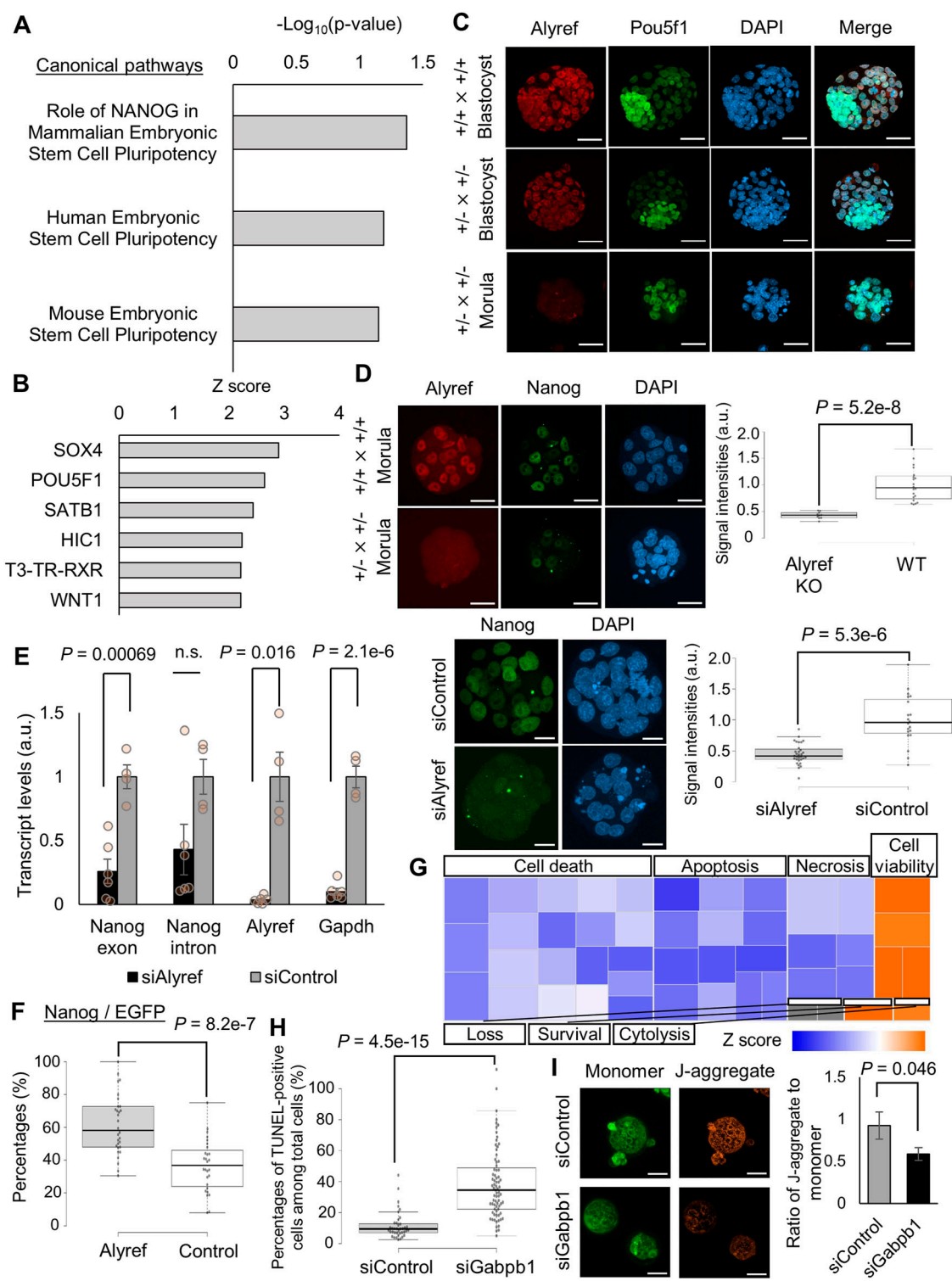

**Figure 6. Molecular mechanisms related to the developmental arrest of *Alyref*$^{-/-}$ and *Gabpb1*$^{-/-}$ embryos.**
**(A)** Canonical pathways predicted by IPA using differentially expressed genes between knockout and WT (and/or heterozygous) morulae (KOA-M1-2 versus A-M1-3). Terms related to mammalian pluripotency are extracted from the identified pathway list (Appendix Table S1). **(B)** Upstream regulators responsible for misexpression in *Alyref*$^{-/-}$ embryos were identified by IPA. Activation Z scores are shown. **(C)** Immunostaining of Alyref and Pou5f1 in in vitro fertilized (IVF) embryos derived from different combinations of crossing. Embryos were collected at 96 hpi. Some of the arrested morula embryos were devoid of Alyref signals (*Alyref*$^{+/-}$ × *Alyref*$^{+/-}$). DNA was stained by DAPI. N = 3. n = (*Alyref*$^{+/+}$ × *Alyref*$^{+/+}$): 72 and (*Alyref*$^{+/-}$ × *Alyref*$^{+/-}$): 95. Scale bars = 50 μm. **(D)** Immunostaining of Alyref and Nanog in IVF embryos derived from different combinations of crossing (top panels) and those injected with *Alyref* or control siRNAs (bottom panels). Box plots (right) indicate signal intensities of Nanog in IVF embryos. Embryos were collected at 72 hpi. Morula embryos devoid of Alyref signals showed weak Nanog expression. WT represents wildtype embryos. N = 3. n = siAlyref: 31,

cumulus cells, fertilized embryos, and SCNT embryos at the two-cell stage, compared with the Dux cluster region that is known to be marked with abundant H3K9me3 in cumulus cells and SCNT embryos (Yang et al, 2021) (Fig S5A). In addition, both *Alyref* and *Gabpb1* were not located around aberrantly low acetylated regions, which correlate with H3K9me3-enriched reprogramming-resistant regions (Yang et al, 2021). Nevertheless, very weak signals of H3K9me3 were detected around transcription start sites of *Alyref* and *Gabpb1* (Fig S5A, red highlighted regions), and we then tried to lower H3K9me3 levels in cumulus cells to see the effect on expression of *Alyref* and *Gabpb1*. siRNA treatment against Suv39h1/2, main regulators for catalyzing H3K9me3 (Matoba et al, 2014), successfully inhibited their expression and reduced H3K9me3 levels in cumulus cells (Fig S5B–D). However, transcriptional up-regulation of *Alyref* and *Gabpb1* was not observed in cumulus cells even after lowing H3K9me3 (Fig S5C). It is noteworthy that both *Alyref* and *Gabpb1* were expressed in donor cumulus cells even without interfering with H3K9me3 (Fig S5C). Taken together, the repressed chromatin state by H3K9me3 in donor cumulus cells cannot explain silencing of *Alyref* and *Gabpb1* in SCNT embryos. Therefore, it is most likely that down-regulated expression of *Alyref* and *Gabpb1* in SCNT embryos is indirectly regulated by H3K9me3.

### Preimplantation development of SCNT embryos is rescued by exogenous supplementation of *Alyref* and *Gabpb1*

We asked if the insufficient expression of *Alyref* and *Gabpb1* is a reason for the low developmental ability of SCNT embryos. *EGFP-Alyref* and *EGFP-Gabpb1* mRNA was injected to SCNT embryos reconstructed with cumulus cells and subsequence preimplantation development was examined (Fig 7A, without TSA). Because the developmental rescue is often dose-dependent, different doses of mRNA mixtures of *Alyref* and *Gabpb1* were injected (100, 400, 500, and 600 ng/µl). *Alyref* and *Gabpb1* mRNA injection enhanced preimplantation development of SCNT embryos (Fig 7B) and the most prominent effect was observed when 400 ng/µl of *Alyref* and *Gabpb1* mRNA (each 200 ng/µl) were injected (Fig 7B and C). As shown in Fig S1D, *Alyref* and *Gabpb1* are activated by removing H3K9me3 with Kdm4d or lowering it with VC. We therefore hypothesized that the positive effect of VC on reprogramming in SCNT embryos can be substituted by the supplementation of *Alyref* and *Gabpb1* mRNA. VC's effect is most prominent when combined with TSA (Miyamoto et al, 2017), and therefore, we injected 400 ng/µl of *Alyref* and *Gabpb1* mRNA to SCNT embryos treated with TSA (Fig 7A, with TSA). Strikingly, 87% of SCNT embryos developed to the

blastocyst stage after *Alyref* and *Gabpb1* mRNA injection (Fig 7D). Finally, we tested if *Alyref* and *Gabpb1* mRNA injection can support efficient development of SCNT embryos to term. SCNT embryos were treated with TSA, together with or without *Alyref* and *Gabpb1* mRNA injection, but significant improvement of the live offspring rate was not observed by supplementation of *Alyref* and *Gabpb1* mRNA (Fig 7E). These results suggest that incomplete activation of *Alyref* and *Gabpb1* is closely linked to preimplantation arrest of SCNT embryos, but restoration of their expression is not sufficient for supporting full-term development of SCNT embryos.

### *Klf16*, repressed in SCNT embryos, can activate *Alyref* and *Gabpb1* at the two-cell stage

Expression of *Alyref* and *Gabpb1* is not repressed by H3K9me3 in donor cumulus cells (Fig S5C). However, their expression is up-regulated in SCNT embryos in which H3K9me3 levels were lowered at the two-cell stage (Figs 2C and S1D) (Matoba et al, 2014; Miyamoto et al, 2017). One possible mechanism of this could be that reprogramming-resistant H3K9me3-repressed genes in SCNT embryos contain upstream transcriptional activators for *Alyref* and *Gabpb1*. We therefore explored transcriptional regulators that are up-regulated after removing the H3K9me3 mark (Matoba et al, 2014; Yang et al, 2021), and that can activate *Alyref* and *Gabpb1* in SCNT embryos. Firstly, gene regulatory regions of *Alyref* and *Gabpb1* were predicted by taking advantage of ENCODE registry of candidate cis-regulatory elements (The ENCODE Project Consortium et al, 2020) and candidate transcription factors that can bind to the predicted gene regulatory regions were listed from JASPAR transcription factor binding site database (Castro-Mondragon et al, 2022). Among the candidate transcription factors, we searched for those that were up-regulated after overexpression of H3K9me3 demethylases in SCNT embryos (Matoba et al, 2014; Yang et al, 2021), leading to the identification of *Klf16* and *Tfap2c* (Fig 8A and B). Down-regulation of *Klf16* transcripts was observed in SCNT embryos at the two-cell stage, as compared with IVF embryos (Fig 8C). Supplementation of 20 ng/µl *Klf16* mRNA in SCNT embryos significantly up-regulated *Alyref* and *Gabpb1* transcripts at the late two-cell stage (Fig 8D, *Alyref*: 1.65-fold and *Gabpb1*: 1.82-fold increases). The magnitude of transcriptional up-regulation in *Klf16*-supplemented SCNT embryos is not as much as that in TSA-VC–treated SCNT embryos (Fig 2C, *Alyref*: 3.71-fold and *Gabpb1*: 2.23-fold increases), suggesting that an additional factor for fully

siControl: 20, Alyref KO: 10 and WT: 20. Scale bars = 50 µm (top embryos from different combinations of crossing), 20 µm (bottom siRNA-injected embryos). **(E)** qRT-PCR analyses of morula embryos at 72 hpi injected with *Alyref* (black) or control siRNA (grey). Relative transcript levels to control samples are shown. siAlyref: 6 and siControl: 4 biological replicates. **(F)** Percentages of Nanog-positive cells among EGFP-Alyref–positive or control H2B-EGFP–positive cells in blastocysts. *EGFP-Alyref* or *histone H2B-EGFP* (control) mRNA was injected into one blastomere of a two-cell stage embryo and Nanog-positive cells were counted at 96 hpi. N = 3. n = *Alyref*: 30 and control: 28. **(G)** Genes related to the cellular functions were misregulated in *Gabpb1*$^{-/-}$ embryos, as revealed by IPA. A heatmap of activation Z scores is shown. **(H)** Percentages of apoptotic cells in IVF embryos injected with *Gabpb1* or control siRNA at 96 hpi, as detected by the TdT-mediated dUTP Nick End Labeling assay. N = 3. n = siGabpb1: 75 and siControl: 39. **(I)** Mitochondrial membrane potential in IVF embryos injected with *Gabpb1* or control siRNA was examined at 96 hpi using 5,5′,6,6′-tetrachloro-1,1′,3,3′-tetraethylbenzimidazolyl carbocyanine iodide (JC-1). JC-1 forms J-aggregates (orange) when it enters the mitochondria. In contrast, JC-1 remains monomeric in cells with low mitochondrial membrane potential (green). The right graph shows mitochondrial membrane potential as a ratio of the average orange fluorescence (high potential) relative to average green fluorescence (low potential) in the injected embryos. N = 3. n = siGabpb1: 13 and siControl: 7. Data information: bars represent mean ± SEM. N number refers to independent experiments. n number refers to the numbers of embryos used for each treatment. n.s. represents not significant. Statistical significance was determined by two-sided F- and t tests.

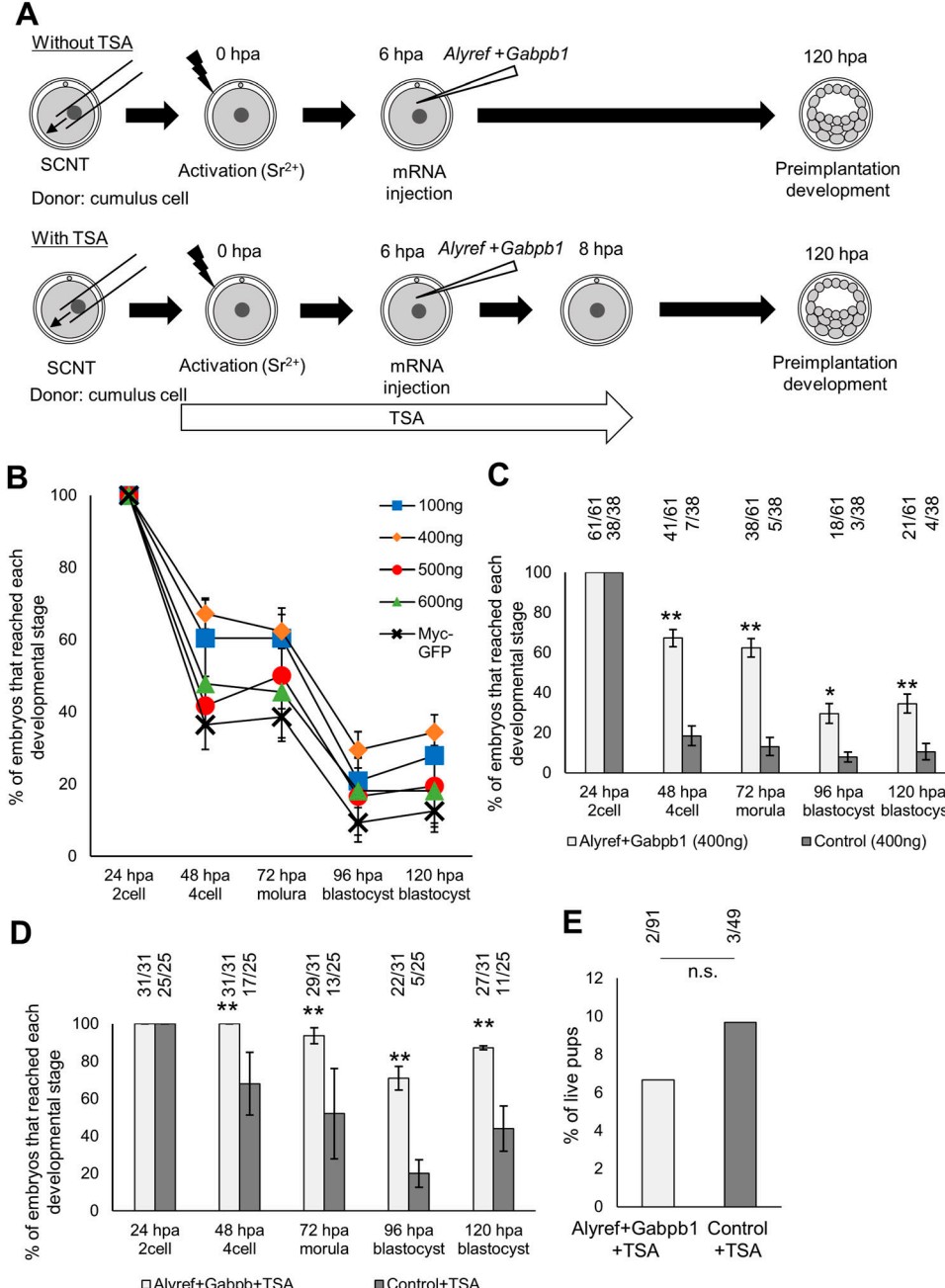

**Figure 7. Developmental rescue of somatic cell nuclear transfer (SCNT) embryos by the supplementation of Alyref and Gabpb1.** **(A)** Experimental design to examine the effect of *Alyref* and *Gabpb1* mRNA injection on the development of SCNT embryos with or without trichostatin A (TSA) treatment. hpa, hours post activation. **(B)** Preimplantation development of SCNT embryos injected with various different concentrations of *EGFP-Alyref* and *EGFP-Gabpb1* mRNA (100–600 ng/µl). *Myc-tag-GFP* mRNA (100–600 ng/µl) was injected as a control. *N* = 4. *n* = 100 ng/µl: 43, 400 ng/µl: 30, 500 ng/µl: 36, 600 ng/µl: 44, Myc-tag-GFP: 96. **(C)** Preimplantation development of SCNT embryos injected with 400 ng/µl of *EGFP-Alyref* and *EGFP-Gabpb1* mRNA or *Myc-tag-GFP* mRNA as a control. The numbers of embryos used for experiments and that reached each developmental stage are indicated. *N* = 4. **(D)** Preimplantation development of TSA-treated SCNT embryos injected with 400 ng/µl of *EGFP-Alyref* and *EGFP-Gabpb1* mRNA or *Myc-tag-GFP* mRNA as a control. The numbers of embryos used for experiments and that reached each developmental stage are indicated. *N* = 3. **(E)** The live offspring rate of TSA-treated SCNT embryos injected with 400 ng/µl of *EGFP-Alyref* and *EGFP-Gabpb1* mRNA or *Myc-tag-GFP* mRNA as a control. The numbers of embryos transferred to foster mothers and full-term live pups are indicated. *N* = 4. *N* number refers to independent injection experiments. *n* number refers to the number of embryos used for each treatment. **(C, D, E)** *$P < 0.05$, **$P < 0.01$, determined by chi-square test for (C, D) and Fisher's exact test for (E). n.s. represents not significant.

activating *Alyref* and *Gabpb1* likely exists. We then asked whether *Tfap2c* can boost *Alyref* and *Gabpb1* expression when combined with the *Klf16* treatment. However, the combinational treatment of *Klf16* and *Tfap2c* mRNA overexpression diminished the positive effect of *Klf16* on *Alyref* and *Gabpb1* activation (Fig 8E). Therefore, we focused on *Klf16* to examine its effect on development of SCNT embryos. *Klf16* overexpression (20 ng/µl) in SCNT embryos alone did not significantly increase the development of SCNT embryos (Fig 8F). This may be because of the partial rescue of *Alyref* and *Gabpb1* expression after supplementation of *Klf16* (Fig 8D). As the positive effect of *Alyref* and *Gabpb1* up-regulation

on preimplantation development is enhanced by the treatment with TSA (Fig 7D), *Klf16*-supplemented SCNT embryos were incubated in TSA-containing medium. *Klf16* overexpression together with TSA significantly enhanced the development of SCNT embryos to the blastocyst stage, as compared with the TSA-treated control SCNT embryos (Fig 8G; 49.7% versus 26.7%). However, *Klf16* overexpression with TSA did not support the efficient full-term development (Fig 8H). In summary, *Alyref* and *Gabpb1* are repressed in SCNT embryos, at least partially, because of the H3K9me3-mediated incomplete activation of *Klf16*.

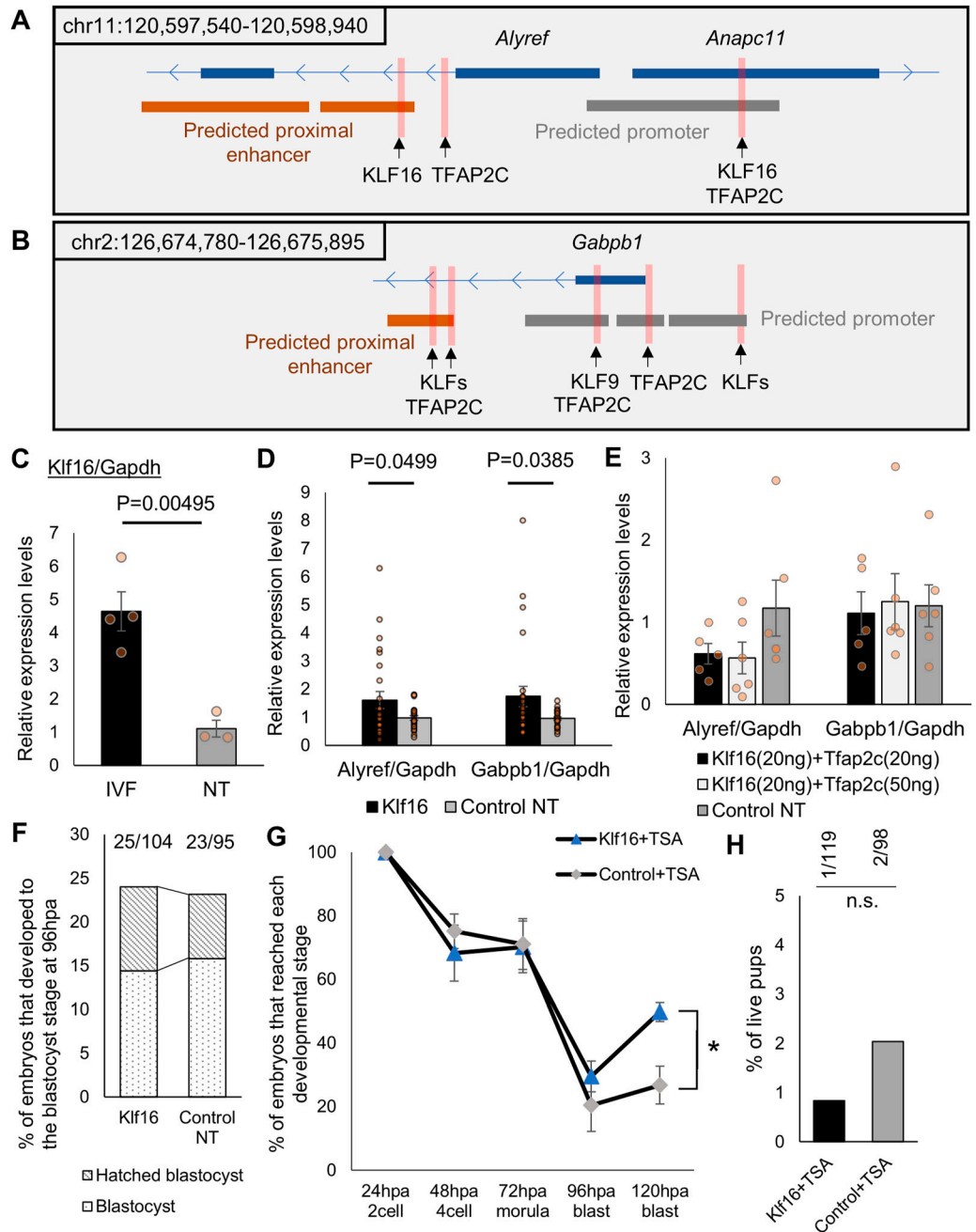

**Figure 8.  Supplementation of *Klf16* up-regulates *Alyref* and *Gabpb1* and improves preimplantation development of somatic cell nuclear transfer (SCNT) embryos.**
**(A, B)** Schematic diagrams show genomic regions around mouse *Alyref* (A) or *Gabpb1* (B). Predicted gene regulatory regions from ENCODE registry of candidate cis-regulatory elements (The ENCODE Project Consortium et al, 2020) are indicated; predicted promoters: grey; predicted proximal enhancers: orange. Pink highlighted areas indicate predicted binding sites for transcription factors. Information was obtained from JASPAR transcription factor binding site database (Castro-Mondragon et al, 2022). **(C)** qRT-PCR analyses of in vitro fertilized embryos at 28 hpi (black) and two-cell SCNT embryos at 28 hpa (grey). Relative transcript levels of *Klf16* to *Gapdh* were measured. In vitro fertilized: 4 and NT: 3 biological replicates. **(D)** qRT-PCR analyses of two-cell SCNT embryos at 28 hpa injected with *Klf16-HA* (black) or control *Myc-tag-GFP* mRNA (grey). Relative transcript levels of *Alyref* or *Gabpb1* to *Gapdh* were measured. *Klf16* NT: 26 and control NT: 25 biological replicates. **(E)** qRT-PCR analyses of two-cell SCNT embryos at 28 hpa injected with *Klf16-HA* (20 ng/μl) with *Tfap2c-HA* (20 ng/μl) (black), *Klf16-HA* (20 ng/μl) with *Tfap2c-HA* (50 ng/μl) (light grey) or control *Myc-tag-GFP* mRNA (grey). Relative transcript levels of *Alyref* or *Gabpb1* to *Gapdh* were measured. *Klf16* (20 ng) + *Tfap2c* (20 ng) NT: 5, *Klf16* (20 ng) + *Tfap2c* (50 ng) NT: 6 and control NT: 6 biological replicates. **(F)** Preimplantation development of SCNT embryos injected with *Klf16-HA* mRNA (20 ng/μl). *Myc-tag-GFP* mRNA was injected as a control. The number of embryos that reached the blastocyst stage at 96 hpa is shown above the bars. N = 9. n = *Klf16*: 104, *Myc-tag-GFP*: 95. **(G)** Preimplantation development of trichostatin A (TSA)-treated SCNT embryos injected with *Klf16-HA* mRNA (20 ng/μl) or *Myc-tag-GFP* mRNA as a control. N = 4. n = *Klf16*+TSA: 50, control *Myc-tag-GFP* + TSA: 41. **(H)** The live offspring rate of TSA-treated SCNT embryos injected with 20 ng/μl of *Klf16-HA* mRNA or *Myc-tag-GFP* mRNA as a control. The numbers of embryos transferred to foster mothers and full-term live pups are indicated. N = 2. Data information: bars represent mean ± SEM. N number refers to independent injection experiments. n number refers to the numbers of embryos used for each treatment. **(C, D, E, F, G, H)** *P < 0.05, determined by chi-square test for (G) and two-sided F- and t tests for (C, D). Statistical differences were analyzed by ANOVA for (E), chi-square test for (F), and fisher's exact test for (H) and did not show significant differences. n.s. represents not significant.

# Discussion

Tremendous efforts have been made to improve the developmental potential of cloned embryos. Recent studies have revealed that the alteration of epigenetic states in SCNT embryos greatly enhances their development to term. Especially, the removal of the repressive H3K9me3 mark is key for development. However, it is still unclear how the reduced H3K9me3 helps in development of SCNT embryos. One assumption is that genes necessary for embryonic development are adequately activated upon the removal of H3K9me3 because the overexpression of H3K9me3 demethylase Kdm4d up-regulated the expression of many zygotically activated genes (Matoba et al, 2014). In line with this concept, our study demonstrates that expression of developmentally essential *Alyref* and *Gapbp1* is normally inhibited in SCNT embryos at the time of ZGA, and the exogenous supplementation of such mRNA can rescue the development of SCNT embryos. Furthermore, H3K9me3-repressed genes contain *Klf16*, a candidate upstream regulator for *Alyref* and *Gapbp1*, and overexpression of *Klf16* can up-regulate *Alyref* and *Gapbp1*. Thus, our article provides mechanistic insight into somatic cell nuclear reprogramming in SCNT embryos and further represents a valid approach to identifying developmentally required genes.

Our siRNA screening identified two developmentally important genes *Alyref* and *Gabpb1*. Embryonic lethal phenotype till E12.5 has been reported after knocking out *Gapbp1* (Xue et al, 2008), but its role for preimplantation development has not been investigated. Our study shows that knockout of *Gabpb1* induces the abnormal gene expression pattern, especially to genes related to antioxidation and cellular viability. GABPB1 forms a tetrameric complex and activates the transcription of various genes, including the antioxidant genes and genes involved in mitochondrial biogenesis (Usmanova et al, 2011; Ryu et al, 2014), suggesting that defective expression of key metabolic genes may lead to dysfunction of the mitochondria and apoptosis around the morula stage. Even the embryos that developed to blastocysts after *Gabpb1* knockdown underwent apoptosis ($P < 0.05$, F- and $t$ tests), in good agreement with the embryonic lethal phenotype (Xue et al, 2008). In the case of *Alyref*, all embryos were arrested at the morula stage after its knockout. Alyref has been shown as a reader protein for 5-methylcytosine (5mC) RNA and involved in the processing and export of such RNA (Yang et al, 2017). 5mC modification facilitates the mRNA stability during maternal-to-zygotic transition in zebrafish (Yang et al, 2019). In our study, the inhibition of *Alyref* caused insufficient expression of Nanog protein presumably by affecting the processing of *Nanog* mRNA (Fig 6E). It is also noteworthy that the overexpression of Alyref in one of the blastomeres at the two-cell stage enhanced percentages of Nanog-positive cells at 96 hpi (Fig 6F). Our studies thus suggest that *Alyref* accelerates cell lineage contribution to ICM through regulation of *Nanog*. Interestingly, insufficient expression of key pluripotency genes such as *Pou5f1* and *Nanog* in SCNT embryos is often observed (Boiani et al, 2002; Mallol et al, 2015). Impaired activation of *Alyref* in SCNT embryos might be related to this phenotypic defect of SCNT embryos.

Repressive histone marks serve as a barrier for somatic cell nuclear reprogramming. *Alyref* and *Gabpb1* were identified as genes

up-regulated upon the removal of H3K9me3 and those by VC treatment that also lowers H3K9me3 levels (Figs 2C and S1D); otherwise repressed in SCNT embryos (Fig 2). Our results suggest that *Alyref* and *Gabpb1* are not up-regulated after lowering the H3K9me3 levels in donor cumulus cells (Fig S5). Rather, *Alyref* and *Gabpb1* are abundantly expressed in cumulus cells without altering the H3K9me3 mark. In line with this finding, ChIP-seq analysis of H3K9me3 using SCNT embryos (Liu et al, 2016) also suggests no obvious accumulation of H3K9me3 around *Alyref* and *Gabpb1* at the two-cell stage. Taken together, the expression of *Alyref* and *Gabpb1* is not directly inhibited by H3K9me3. We then assume that transcriptional activators of *Alyref* and *Gabpb1* are suppressed by H3K9me3, eventually resulting in the repression of *Alyref* and *Gabpb1* in SCNT embryos. Activators for *Alyref* and *Gabpb1* should be expressed just before or at the time of major ZGA in fertilized embryos, but not in SCNT embryos because of the high level of H3K9me3. One candidate would be *Dux*, which has been shown as a major regulator for ZGA (Yang et al, 2021). However, both *Alyref* and *Gabpb1* were not included in up-regulated genes by *Dux* overexpression in SCNT embryos (Yang et al, 2021). Therefore, we searched for other activators, which are under the control of H3K9me3-mediated repression and possess binding sites at the gene regulatory regions of *Alyref* and *Gabpb1*. Finally, we showed that *Klf16*, which has been shown as a *Kdm4b*-regulated gene (Yang et al, 2021), can up-regulate *Alyref* and *Gabpb1* in SCNT embryos at the two-cell stage. Thus, our study suggests that the abnormally silenced ZGA genes in SCNT embryos include those indirectly repressed by H3K9me3 because of the incomplete activation of transcriptional activators.

As discussed, *Alyref* and *Gabpb1* are indirectly regulated by H3K9me3; it would therefore be difficult to alter chromatin states of donor cells for the expression of *Alyref* and *Gabpb1* in SCNT embryos. Instead, supplementation of *Alyref* and *Gabpb1* mRNA to SCNT embryos by microinjection would rescue their expression and we have shown that the preimplantation development of SCNT embryos can be enhanced by injecting *Alyref* and *Gabpb1* mRNA (Fig 7). These results suggest that incomplete activation of *Alyref* and *Gabpb1* is at least partially responsible for the low developmental potential of SCNT embryos. However, considering that the remarkable rescue of development to blastocysts is only observed in combination with TSA (Fig 7D), insufficient expression of *Alyref* and *Gabpb1* cannot fully explain the H3K9me3-caused low developmental ability of SCNT embryos. Our embryo transfer experiments also support this notion because highly efficient offspring rate, as is observed in the case of H3K9me3 removal and TSA + VC, was not seen even after the combinational treatment with TSA and *Alyref* + *Gabpb1*. For the purpose of improving the cloning efficiency, supplementation of transcriptional activators that can simultaneously regulate critical ZGA genes would be appropriate as shown in the case of Dux (Yang et al, 2021). Future investigations are needed to further refine the optimal mixture of key transcriptional regulators to support efficient development of SCNT embryos, which may also lead to the comprehensive understanding of the H3K9me3-mediated barrier for nuclear reprogramming to the totipotent state.

Based on these observations, we propose the nature of the low developmental ability of SCNT. Firstly, the condensed chromatin

state of somatic nuclei prevents the access of maternal reprogramming factors. Global relaxation of chromatin by increasing histone acetylation, such as by TSA, overcomes this barrier. Secondly, reprogramming-resistant H3K9me3 marks and the aberrantly acetylated state in somatic chromatin prevent activation of developmental genes. Our study demonstrates that *Alyref* and *Gabpb1* are one of the key genes that are normally inhibited at the time of ZGA in SCNT embryos by the effect of reprogramming-resistant H3K9me3. In this case, H3K9me3 represses *Klf16* that is partially responsible for *Alyref* and *Gabpb1* expression. Thirdly, the lack of H3K27me3-mediated imprinting in somatic chromatin results in misregulation of miRNA clusters, which are closely related to the proper placenta formation (Inoue et al, 2020; Wang et al, 2020). Taken together, it is intriguing to know that the impaired development of SCNT embryos can be rescued by several key factors at different developmental stages, indicative of progressive reprogramming in SCNT embryos. Our study sheds light on the previously unexplored genes that are important for somatic nuclei to obtain totipotency after nuclear transfer. Furthermore, our results provide a proof of concept that impaired preimplantation development of SCNT embryos is caused by the incomplete activation of developmentally necessary genes at ZGA.

## Materials and Methods

### Animals

Mice (C57BL/6, DBA/2, MCH [ICR] strains and B6D2F1 [C57BL/6J × DBA/2N]) at 8–10 wk of age were purchased from CLEA Japan or Japan SLC and maintained in light-controlled, air-conditioned rooms. B6D2F1 mice were used for IVF. This study was carried out in strict accordance with the recommendations in the Guidelines of Kindai University for the Care and Use of Laboratory Animals. Experimental protocols were approved by the Committee on the Ethics of Animal experiments of Kindai University (Permit Number: KABT-31-003). All mice were euthanized by cervical dislocation and all efforts were made to minimize suffering and to reduce the number of animals used in the present study.

KO mice were generated on the B6D2F1 background with CRISPR/Cas9-mediated genome editing as reported before (Mashiko et al, 2013; Oji et al, 2016). Genotyping of KO mice was performed as described in Figs 3 and 4. PCR products were directly used for Sanger sequencing. Sequences of gRNA and primers are listed in Appendix Table S2. Animal experiments for generating KO mice were approved by the Animal Care and Use Committee of the Research Institute for Microbial Diseases, Osaka University (Permit Number: H30-01-1 and R03-01-0).

### Cell culture and transfection

Cumulus cells were collected from cumulus–oocyte complexes obtained from female B6D2F1 mice by treating with hyaluronidase (H3884; Merck). The collected cumulus cells were seeded in a 24 well and cultured in the medium containing DMEM (11054001; Thermo Fisher Scientific), sodium pyruvate, 10% FBS, and penicillin/ streptomycin at 37°C under 5% $CO_2$ in air before transfection. siRNA transfection for knockdown of Suv39h1/2 was performed as reported previously (Matoba et al, 2014). Briefly, 5 pM siRNAs (Suv39h1 siRNA [s74607; Thermo Fisher Scientific], Suv39h2 siRNA [s82300; Thermo Fisher Scientific]) were transfected into cumulus cells at the 80% confluency using Lipofectamine RNAiMAX Transfection Reagent (13778030; Thermo Fisher Scientific) (day 1). 24 h after the first transfection, the culture medium was changed to a fresh one. On day 4, transfection was repeated as described above. Forty-eight h after the second transfection (day 6), the cumulus cells were subjected to Western blot and qRT-PCR analyses.

### IVF and embryo culture

Collection of spermatozoa, oocytes, and zygotes were performed as described in previous studies (Okuno et al, 2020). Briefly, spermatozoa were collected from the cauda epididymis of B6D2F1 fertile male mice (>8 wk of age). The sperm suspension was incubated in human tubal fluid (HTF) medium (ARK Resource) for 1.5 h to allow for capacitation at 37°C under 5% $CO_2$ in air. Oocytes were collected from the excised oviducts of B6D2F1 female mice (>8 wk of age) that were superovulated with pregnant mare serum gonadotropin (PMSG; Serotropin, ASKA Pharmaceutical Co.) and 48 h later, human chorionic gonadotropin (hCG; ASKA Pharmaceutical Co.). Cumulus–oocyte complexes were recovered into pre-equilibrated HTF medium. The sperm suspension was added to the oocyte cultures and morphologically normal zygotes were collected 2 h post-insemination (hpi). The zygotes were cultured in potassium simplex-optimized medium KSOMaa (ARK Resource) at 37°C under 5% $CO_2$ in air.

### SCNT

SCNT was carried out as described previously (Miyamoto et al, 2017). Briefly, enucleation of denuded MII oocytes was performed in drops of HCZB containing 5 μg/ml cytochalasin B (Sigma-Aldrich). After enucleation, a donor cell in HCZB with 6% dBSA was placed in the perivitelline space of an enucleated oocyte together with HVJ-E (GenomeONE-CF; Ishihara Sangyo) by tightly attaching the donor cell to the enucleated oocyte; the oocyte was then cultured in KSOMaa for 1 h at 37°C in air containing 5% $CO_2$, during which time, it fused with the donor cell. The reconstructed oocytes were activated by the incubation for 6 h in 5 mM $SrCl_2$ and 2 mM EGTA-containing KSOMaa supplemented with 5 μg/ml cytochalasin B, referred to as the activation medium (Kishigami & Wakayama, 2007), at 37°C in air containing 5% $CO_2$. For the treatment with TSA (T8552; Sigma-Aldrich), NT embryos were treated with 50 nM TSA for 8 h from the commencement of activation. The activated NT embryos were cultured at 37°C in air containing 5% $CO_2$ in KSOMaa.

### Embryo transfer

Pseudopregnant ICR females were prepared as recipients for embryo transfer by mating with sterile ICR males. SCNT embryos at the two-cell stage were transferred to the oviducts of the pseudopregnant females at 0.5 d post-coitus (dpc). Cesarean section and uterine analysis of implantation sites were performed in all

recipients at 19.5 dpc. If available, lactating foster mothers were used to raise live pups.

## Plasmids

Constructs were produced either using a gateway system (Thermo Fisher Scientific) or in-fusion cloning (Clontech). Myc-tag-GFP was subcloned into the pCS2 vector (Miyamoto et al, 2011) for mRNA production as described previously (Baarlink et al, 2017). pCS2-myc-GFP and pcDNA3.1-Histone H2B-EGFP were reported previously (Baarlink et al, 2017; Hatano et al, 2022). To obtain the *Gabpb1*, *Klf16*, and *Tfap2c* mRNA-producing vectors for injection experiments, we first generated the pENTR/D-TOPO-Gabpb1 entry vector. After clonase-triggered recombination reaction, pCS2-EGFP-Gabpb1, pCS2-Klf16-HA, and pCS2-Tfap2c-HA were generated. *Gabpb1* insert was amplified from cDNA of mouse blastocyst embryos by using oligos GW_Gabpb1_N_F and GW_Gabpb1_N_R (Appendix Table S2). *Klf16* and *Tfap2c* inserts were amplified from cDNA of two-cell embryos by using oligos GW_mKlf16_C_F, GW_mKlf16_C_R, GW_mTfap2c_C_F, and GW_mTfap2c_C_R (Appendix Table S2). pCS2-EGFP-Alyref was produced by in-fusion cloning. For this subcloning, *Alyref* insert was amplified from the pCMV6-Entry-Alyref-Myc-DDK-tagged plasmid (MR220236; OliGene Technologies) using IF-Alyref-N_Fw and IF-Alyref-N_Rv3 primers. In-fusion reaction was performed after amplification of the linearized pCS2 vector, which was amplified by PCR using IF-Alyref-vectorF and IF-Alyref-vectorR primers. All primers used for cloning are listed in Appendix Table S2.

## mRNA production

mRNAs were prepared from pCS2 vectors using mMESSAGE mMACHINE SP6 Transcription Kit (AM1340 or AM1344; Thermo Fisher Scientific). Briefly, to produce linearized vectors, ~5-$\mu$g plasmids were digested overnight, with appropriate restriction enzymes. In the case of pCS2 vectors, mRNAs produced from the SP6 promoter were subjected to the addition of polyA tails (AM1350; Thermo Fisher Scientific). Produced mRNAs were purified using an Rneasy Mini Kit (74106; QIAGEN).

## mRNA injection

After activation, NT embryos were collected at 6–7 hours post-activation (hpa) for mRNA injection. NT embryos were then washed with KSOMaa and kept in drops of CZB-HEPES medium for injection. mRNAs were injected using a piezo manipulator (Prime Tech). The final concentrations of the injected mRNA are as follows; 50–300 ng/$\mu$l *EGFP-Alyref* mixed with 50–300 ng/$\mu$l *EGFP-Gabpb1*, 100–600 ng *Myc-tag-GFP* for Fig 7B, 200 ng/$\mu$l *EGFP-Alyref* mixed with 200 ng/$\mu$l *EGFP-Gabpb1* for Fig 7C–E, 20 ng *Klf16* for Fig 8D–H, 20 or 50 ng *Tfap2c* for Fig 8E, 20–400 ng *Myc-tag-GFP* for Figs 7C–E and 8D–H. After injection, embryos were cultured in KSOMaa medium at 37°C in air containing 5% $CO_2$. For the TSA treatment, 50 nM TSA was supplemented in the KSOMaa medium and NT embryos were cultured in the TSA-containing medium. After injecting mRNA, embryos were immediately transferred to the inhibitor-containing medium.

For Fig 6F, IVF embryos were collected at 24–27 hpi for mRNA injection. Injection to two-cell embryos was performed in drops of CZB-HEPES medium. mRNAs were injected into one blastomere of a two-cell embryo using a piezo manipulator. The final concentrations of injected mRNA are as follows: 50 ng/$\mu$l of *EGFP-Alyref* or 50 ng of *Histone H2B-EGFP* in Fig 6F. After injection, embryos were cultured in the KSOMaa medium at 37°C in air containing 5% $CO_2$.

## TdT-mediated dUTP Nick End Labeling (TUNEL) assay

For counting apoptotic cells after *Gabpb1* knockdown or knockout, In Situ Cell Death Detection Kit (11684795910; Roche) was used according to the vendor's instruction. Briefly, embryos at 96 hpi were fixed with 4% PFA/PBS at RT for 20 min, and were washed by 3 mg/ml PVP/PBS for three times. Zona pellucida was removed in Tyrode's solution (T1788; Sigma-Aldrich) and the zona-free embryos were washed by PVP/PBS for three times. Then, the embryos were treated with 0.5% Triton X-100 in PBS at RT for 20 min. The TUNEL reaction mixture (5-$\mu$l enzyme solution and the 45-$\mu$l label solution) was prepared immediately before use, and the embryos were incubated in 50 $\mu$l of the TUNEL reaction mixture at 37°C. The samples were washed with PVP/PBS three times and then mounted on slides using VECTA-SHIELD Mounting Medium containing DAPI. The fluorescence signals were observed using an LSM800 microscope (Carl Zeiss), equipped with a laser module (405/488/561/640 nm) and GaAsP detector, using the same contrast, brightness, and exposure settings within the same experiments. Z-slice thickness was determined by using the optimal interval function in the ZEN software.

## Mitochondrial membrane potential analysis

The membrane-sensitive dye JC-1 (5,5',6,6'-tetrachloro-1,1',3,3'-tetraethylbenzimidazolyl carbocyanine iodide) was used to detect mitochondrial membrane potential ($\Delta\Psi$ m) according to the vendor's instruction with some modifications (10009172; Cayman Chemical Company). Briefly, the JC-1 stock solution was diluted (x40) in the KSOMaa medium (KSOM + JC1). Embryos injected with siRNAs against *Gabpb1* or control siRNAs were incubated in KSOM + JC1 for 30 min at 37°C in air containing 5% $CO_2$. After the incubation, embryos were washed three times in phenol red-free KSOMaa medium containing 0.00025% PVA (KSOM + PVA). Then, embryos were transferred to drops of KSOM + PVA on a glass-bottom dish (Matsunami) and placed in an incubation chamber stage (Tokai Hit) at 37°C under 5% $CO_2$ in air for live-cell imaging. The fluorescence signals were observed using an LSM800 confocal microscope, equipped with a laser module (405/488/561/640 nm) and GaAsP detector, using the same contrast, brightness, and exposure settings (3 $\mu$m interval). For detecting J-aggregates and monomers, the rhodamine and FITC filter settings were used, respectively. Membrane potential is expressed as a ratio of the average rhodamine fluorescence (J-aggregates) relative to the average FITC fluorescence (monomers) in embryo areas.

 **Life Science Alliance**

## Immunofluorescence staining

Embryos were fixed in 4% PFA/PBS at RT for 15 min, and were washed by 3 mg/ml PVP/PBS for three times. Zona pellucida was removed in Tyrode's solution and the zona-free embryos were washed by 3 mg/ml PVP/PBS for three times. Samples were next treated with 0.25% Triton X-100 in 3 mg/ml PVP/PBS at RT for 30 min. Blocking was performed in 3% BSA/PBS with 0.01% Tween 20 for 1 h at RT, then embryos were incubated with primary antibodies diluted in 3% BSA/PBS with 0.01% Tween 20 (1:50; anti-ALY antibody [sc-32311; Santa Cruz Biotechnology], 1:200; GABPB1 antibody [GTX103464; Gene Tex], 1:200; Oct4 [Pou5f1] antibody [09-0023; REPROCELL], 1:100; Oct3/4 [Pou5f1] antibody [sc5279; Santa Cruz Biotechnology], 1:300; Nanog Antibody [RCAB002P-F; REPROCELL]) at 4°C overnight. After three times washes by 1% BSA/PBS with 0.01% Tween 20, samples were further incubated in the dark with Alexa Fluor 488-labeled goat anti-mouse IgG antibody (1:2,000; A11001; Thermo Fisher Scientific), Alexa Fluor 488-labeled goat anti-rabbit IgG antibody (1:2,000; A11008; Thermo Fisher Scientific), Alexa Fluor 594-labeled donkey anti-mouse IgG antibody (1:2,000; A21203; Thermo Fisher Scientific) or Alexa Fluor 594-labeled donkey anti-rabbit IgG antibody (1:2,000; A21207; Thermo Fisher Scientific) at RT for 1 h. The samples were washed with 1% BSA/PBS with 0.01% Tween 20 three times and then mounted on slides using VECTA-SHIELD Mounting Medium containing DAPI. The fluorescence signals were observed using an LSM800 microscope, equipped with a laser module (405/488/561/640 nm) and GaAsP detector, using the same contrast, brightness, and exposure settings within the same experiments. Z-slice thickness was determined by using the optimal interval function in the ZEN software.

## Image analysis

Images were analyzed using the ZEN software. For quantification of Nanog signals in IVF embryos (Fig 6D), all focal planes were merged by maximum intensity projection. Nanog staining did not show nonspecific cytoplasmic signals and it was localized in nuclei. Therefore, the maximum intensity projection using all focal planes was used for quantifying Nanog signals. After maximum intensity projection, intensities of Nanog in nuclei were quantified. Furthermore, background signals were subtracted from the nuclear signals and the average signal intensities of the merged focal planes were calculated by using the "Measure" function in the ZEN software. To quantify nuclear signal intensities of Alyref and Gabpb1 (Fig 2D and E), all focal planes that cover the nuclear region were used for quantifying nuclear signal intensities. In each focal plane, background signals were subtracted from the nuclear signals and the average of all focal planes was calculated by using the "Measure" function in the ZEN software.

For Fig 6F, the injected embryos were cultured till 96 hpi and blastocyst embryos were collected. The collected blastocysts were fixed and stained with Nanog and DAPI. Then, EGFP-positive cells and Nanog-positive ICM cells, together with the total number of cells with DAPI, were counted by the confocal microscopy.

## Western blot

Cumulus cells grown in a 24-well were suspended in the SDS sample buffer (196-11022; Wako), and the lysed samples were subjected to SDS–PAGE (15% poly-acrylamide gel). The gels were transferred onto polyvinylidene difluoride membranes, and the membranes were blocked with 3% BSA in PBS-Tween (PBS-T) solution for 1 h. The blocked membranes were incubated with anti-H3K9me3 (ab8898; Abcam) or anti-ACTB antibodies (A5441; Merck) overnight at 4°C. After washing, the membranes were incubated with HRP-conjugated donkey anti-rabbit IgG secondary antibody (AP182P; Millipore) or IRDye 680RD goat anti-rabbit IgG secondary antibody (925-68071; LI-COR), and visualized using ECL prime Western blotting detection reagents (RPN2232; Amersham) or Odyssey imaging system (LI-COR).

## RNA-seq analysis

Timings for sampling are shown in each corresponding figure. A single-NT embryo at the morula or blastocyst stage was treated with acid Tyrode, followed by three washes with 0.1% BSA/PBS, and was moved into 1 × reaction buffer from SMART-seq v4 Ultra Low Input RNA Kit (Z4888N; Takara). SMART-seq library preparation was performed using SMART-seq v4 Ultra Low Input RNA Kit and Nextera DNA Sample Preparation Kit (FC-131-1024; illumine) according to the vendor's instruction. We followed the previously published protocol for RNA-seq of mouse embryos (Tomikawa et al, 2021).

Briefly, paired-end sequencing (50 + 25 bp) was obtained by the NextSeq platform (Illumina). Raw reads were first subjected to filtering to remove low-quality reads using Trimmomatic (Bolger et al, 2014). Reads of less than 20 bases and unpaired reads were also removed. Furthermore, adaptor, polyA, polyT, and polyG sequences were removed using Trim Galore! (https://www.bioinformatics.babraham.ac.uk/projects/trim_galore/). The sequencing reads were then mapped to the mouse genome (mm10) using STAR (Dobin et al, 2013). Reads on annotated genes were counted using featureCounts (Liao et al, 2014). RNA-seq reads were visualized using Integrative Genome Viewer (Robinson et al, 2011). Fragments per kilobase of exon per million mapped values were calculated from mapped reads by normalizing to the total count and transcript. DEGs, $P < 0.05$ were then identified using DESeq2 (Love et al, 2014). A hierarchical clustering of the read count values was performed using hclust in TCC (unweighted pair group method with arithmetic mean). Gene ontology terms showing overrepresentation of genes that were up- or down-regulated were detected using DAVID tools ($P < 0.05$). Principal-component analysis of the global gene expression profile was performed using scikit-learn and the plots were depicted using plotly. Each gene list was further subjected to an Ingenuity Pathway Analysis (IPA; QIAGEN). Using IPA, enriched canonical pathways, upstream transcriptional regulators, and diseases and biological functions were investigated.

For Fig S1D, Fragments per kilobase of exon per million mapped values were obtained from GSE59073 and GSE95053.

### ChIP-sequencing (ChIP-seq) analysis

The published ChIP-seq dataset (GSE70606) was used for examining H3K9me3 enrichment around *Alyref* and *Gabpb1* in Fig S5A. Sequencing reads were trimmed to remove adapters using trim galore (v0.6.7). The trimmed reads were mapped to mouse genome (UCSC mm10) using Bowtie2 (v2.4.5) with the parameters: -t–no-mixed–no-discordant–no-unal (Langmead & Salzberg, 2012; Yang et al, 2021). Duplicated reads were removed using samtools (v0.6.8) (Tarasov et al, 2015). H3K9me3 signals were calculated using a 50-bp window and normalized to the uniquely mapped fragments using "bam-Coverage" from deepTools (v3.5.1) with the parameters: –normalizeUsing RPKM.

### qRT-PCR

Total RNA was extracted from a pool of three four-cell embryos at 48 hpi for Fig S1A, three two-cell embryos at 28 hpa or 28 hpi for Figs 2C and 8C–E, and three morula embryos at 72 hpi for Fig 6E using PicoPure RNA Isolation Kit (KIT0204; Thermo Fisher Scientific), or from cumulus cells grown in a 24-well using RNeasyMini Kit (74106; QIAGEN) according to the manufacturer's protocol. cDNA was synthesized from the extracted RNA using appropriate RT primers (oligo [dT] primers for Figs 2C, 8C–E, and S1A, and random hexamers for Fig 6E) and SuperScript III Reverse Transcriptase (18080044; Thermo Fisher Scientific). cDNA samples were analyzed in a 7300 Real-Time PCR System (Applied Biosystems) or StepOne Real-Time PCR System (Applied Biosystems). The primer sequences are listed in Appendix Table S2.

### siRNA knockdown in embryos

For investigating a function of candidate genes responsible for acquiring totipotency, specific siRNA was injected to IVF embryos at 6 hpi, and the injected embryos were further cultured in the KSOMaa medium to observe preimplantation development. Pre-designed negative control siRNA (RNAi Inc.) was used as a control. Sequences for siRNA and qRT-PCR primers are listed in Appendix Table S2.

### Statistical analysis

All of the statistical methods for each experiment can be found in the figure legends and in the relevant Materials and Methods section. For Figs 1C, 2C–E, 4D, 6D–F, H, and I, 8C and D, S1A, S2B, and S5C, statistical significance was calculated by two-sided F- and $t$ tests. For Figs 1D, 3C, 4E, 7C and D, 8F and G, and S1B, statistical significance was calculated by chi-square test. For Fig 8E, statistical significance was calculated by ANOVA. For Figs 7E and 8H, statistical significance was calculated by Fisher's exact test. Sample size was estimated based on our preliminary experiments and no statistical method was used. Embryos were randomly selected for image and gene expression analyses. Blinding was applied for the analysis of image data. Embryos that showed abnormal appearance such as degenerated embryos were not subjected for further image and gene expression analyses.

## Data Availability

The datasets produced in this study are available in the following database: RNA-Seq data: Gene Expression Omnibus GSE199874. Reviewer token: mzabeawejrshhkp.

## Supplementary Information

## Acknowledgements

We thank Mr. Kajikuri, Ms. Matsuzawa, and Ms. Kusakabe for their contributions in initiating the project. We thank KK DNAFORM for RNA-seq analyses. We thank Ms. N Backes Kamimura for proofreading. This research was supported by JSPS KAKENHI Grant Numbers JP19H05271, JP19H05751, JP20K21376 to K Miyamoto by The Naito Foundation, Takeda Science Foundation, a Kindai University Research Grant (19-II-1) to K Miyamoto.

### Author Contributions

S Ihashi: data curation, formal analysis, validation, investigation, visualization, and writing—original draft.
M Hamanaka: formal analysis and investigation.
M Kaji: formal analysis and investigation.
R Mori: formal analysis and investigation.
S Nishizaki: validation and investigation.
M Mori: investigation.
Y Imasato: investigation.
K Inoue: investigation.
S Matoba: investigation.
N Ogonuki: investigation.
A Takasu: formal analysis.
M Nakamura: investigation.
K Matsumoto: resources.
M Anzai: investigation and methodology.
A Ogura: supervision and investigation.
M Ikawa: resources, supervision, and methodology.
K Miyamoto: conceptualization, data curation, formal analysis, supervision, funding acquisition, validation, investigation, visualization, methodology, and writing—original draft, review, and editing.

### Conflict of Interest Statement

The authors declare that they have no conflict of interest.

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
