## [Reviewer comments · Life Science Alliance]

Life Science Alliance

Incomplete activation of *Alyref* and *Gabpb1* leads to preimplantation arrest in cloned mouse embryos

Shunya Ihashi, Mizuto Hamanaka, Masaya Kaji, Ryunosuke Mori, Shuntaro Nishizaki, Miki Mori, Yuma Imasato, Kimiko Inoue, Shogo Matoba, Narumi Ogonuki, Atsushi Takasu, Misaki Nakamura, Kazuya Matsumoto, Masayuki Anzai, Atsuo Ogura, Masahito Ikawa, and Kei Miyamoto

DOI: <https://doi.org/10.26508/lsa.202302296>

Corresponding author(s): Kei Miyamoto, Kindai University

Review Timeline:	Submission Date:	2023-07-28
	Editorial Decision:	2023-08-02
	Revision Received:	2023-08-02
	Accepted:	2023-08-04

Scientific Editor: Novella Guidi

Transaction Report:

Please note that the manuscript was previously reviewed at another journal and the reports were taken into account in the decision-making process at *Life Science Alliance*.

Reviews

Referee #1

Report for Author:

Previously, I suggested the authors to address (1) the regulation of Alyref and Gabpb1 expression and (2) the molecular function of Alyref and Gabpb1 in mouse embryos.

Regarding point (1), the authors now provide additional data and somehow conclude that the lower expression of Klf16 leads to the downregulation of Alyref and Gabpb1. However, the data provided are superficial and unconvincing as summarized below.

1. The manuscript does not describe how enhancers in early embryos were defined or how Klf16 and Tfp2c were identified as the candidate molecules. The results based on unbiased analysis are not properly presented.
2. While Klf family genes commonly recognize GC-box elements, which are prevalent at the promoters of many genes, the manuscript does not demonstrate why Klf16 specifically can be considered a candidate molecule. Moreover, additional assays such as reporter assays should be performed to demonstrate the direct effect of Klf16 on the target genes.
3. The overall data in Fig. 8 show that other factors than Klf16 are much more important for the regulation of Alyref or Gabpb1 expression. Thus, the data in this part do not strengthen the conclusion of this paper.

Regarding point (2), in the response letter, the authors state that "As the main focus of the paper is to identify specific genes responsible for the arrest of NT embryos, the suggested experiment is beyond the scope of this paper". However, I believe that describing the molecular function or activity of the identified molecules in mouse embryos is essential to increase the novelty of the findings in this study, and such a rigorous analysis is required for the publication of papers in this journal. I had previously suggested that the authors pursue this direction in my previous comments, but unfortunately, the suggestion has been taken as trivial by the authors.

Referee #2

Report for Author:

The additional data in the revised manuscript have significantly improved the quality of this paper, but I'm not quite satisfied with the response of the following two points. For major point 1, the regulatory relationship between these two genes and H3K9me3 has not been explained clearly. For minor point 4, combination of TSA and Alyref1+Gabpb1 mRNA injection couldn't increase the birth rate of SCNT embryos. This result greatly reduces the significance of this study.

Referee #1 Review

Report for Author:

The comments from the two reviewers were essentially the same, which requested experiments to address molecular mechanisms underlying the authors' findings. I understand that Ihashi et al. have put efforts to respond to comments from the reviewers. However, I feel that the additional data are not sufficient for the publication. I describe some reasons below.

1. There is no direct explanation why *Alyref* and *Gabpb1* are downregulated in SCNT embryos. The authors provide plausible speculation citing some previously published papers from other groups, however, do not sufficiently respond to the comments from the reviewers.
2. The authors now show that expression of *Nanog* rather than *Oct4* seems to be more severely affected by loss of *Alyref* conducting additional immunofluorescence and RT-qPCR on the ICM marker *Nanog*, however, there is no more data than this. If the authors consider that the export of *Nanog* or other mRNA is dysregulated in *Alyref* KO embryos, the possibility should be directly examined. I strongly encourage the authors for future experiments to address the cause rather than the consequence of the phenotype.
3. Line 129: References are not properly indicated.

Referee #2 Review

Report for Author:

The revised manuscript has significantly improved the quality of this paper, but I'm not quite satisfied with the response of the following two points. For major point 1, the regulatory relationship between these two genes and H3K9me3 has not been explained clearly. For minor point 4, combination of TSA and *Alyref1+Gabpb1* mRNA injection couldn't increase the birth rate of SCNT embryos. This result greatly reduces the significance of this study. Given the current challenges of the SCNT field, the most important issue is low developmental potential of SCNT embryos, results in low efficiency of obtaining cloned animals and impedes the prevalence of this technology. Just improves preimplantation development of SCNT embryos lacks of practical value.

Reviewers' comments

Reviewer #1:

Reviewer #1:

The comments from the two reviewers were essentially the same, which requested experiments to address molecular mechanisms underlying the authors' findings. I understand that Ihashi et al. have put efforts to respond to comments from the reviewers. However, I feel that the additional data are not sufficient for the publication in this journal. I describe some reasons below.

1. There is no direct explanation why Alyref and Gabpb1 are downregulated in SCNT embryos. The authors provide plausible speculation citing some previously published papers from other groups, however, do not sufficiently respond to the comments from the reviewers.

Author response: We appreciate the reviewers' comments. We understand that the main reservation, pointed out by both reviewers, is the lack of the mechanistic understanding of how Alyref and Gabpb1 are repressed in SCNT embryos. We now find that Klf16 can upregulate Alyref and Gabpb1 expression in SCNT embryos, but is normally repressed in SCNT embryos (Fig 8). Namely, Alyref and Gabpb1 are repressed in SCNT embryos at least partially due to the H3K9me3-mediated incomplete activation of Klf16 (see Line311-342). Thus, our new data reveal how developmentally important Alyref and Gabpb1 are downregulated in SCNT embryos.

2. The authors now show that expression of Nanog rather than Oct4 seems to be more severely affected by loss of Alyref conducting additional immunofluorescence and RT-qPCR on the ICM marker Nanog, however, there is no more data than this. If the authors consider that the export of Nanog or other mRNA is dysregulated in Alyref KO embryos, the possibility should be directly examined. I strongly encourage the authors for future experiments to address the cause rather than the consequence of the phenotype.

Author response: Thank you for the comments. The suggested experiment is indeed important to pursue the molecular mechanism of how Alyref regulates preimplantation development. As the main focus of the paper is to identify specific genes responsible for the arrest of NT embryos, the suggested experiment is beyond the scope of this paper.

3. Line 129: References are not properly indicated.

Author response: We have added two references. To note, we followed the style of EMBO rep for citing Data ref, and the following dataset were originally listed in the reference list (Line137-138).

Reviewer #2:

The revised manuscript has significantly improved the quality of this paper, but I'm not quite satisfied with the response of the following two points.

For major point 1, the regulatory relationship between these two genes and H3K9me3 has not been explained clearly.

For minor point 4, combination of TSA and Alyref1+Gabpb1 mRNA injection couldn't increase the birth rate of SCNT embryos. This result greatly reduces the significance of this study. Given the current challenges of the SCNT field, the most important issue is low developmental potential of SCNT embryos, results in low efficiency of obtaining cloned animals and impedes the prevalence of this technology. Just improves preimplantation development of SCNT embryos lacks of practical value.

Author response: We appreciate the reviewers' comments. We understand that the main reservation, pointed out by both reviewers, is the lack of the mechanistic understanding of how Alyref and Gabpb1 are repressed in SCNT embryos. We now find that Klf16 can upregulate Alyref and Gabpb1 expression in SCNT embryos, but is normally repressed in SCNT embryos (Fig 8). Namely, Alyref and Gabpb1 are repressed in SCNT embryos at least partially due to the H3K9me3-mediated incomplete activation of Klf16 (see Line311-342). Thus, our new data reveal how developmentally important Alyref and Gabpb1 are downregulated in SCNT embryos.

We also performed embryo transfer experiments. However, Klf16 overexpression was not sufficient to improve the full-term development of SCNT embryos. This result is in line with our previous Alyref and Gabpb1 overexpression experiments (Fig. 7E) and confirms that the Klf16-Alyref/Gabpb1 pathway is responsible for the preimplantation development, but not full-term development.

Thus, we have answered the main reservation, namely the lack of the mechanistic understanding of how Alyref and Gabpb1 are repressed in SCNT embryos. However, we cannot answer the minor point 4 of reviewer 2. From our experiments, it is now clear that the Klf16-Alyref/Gabpb1 pathway plays a role in preimplantation development of SCNT embryos, but not the development after implantation. Therefore, it is difficult to improve the full-term development by manipulating the Klf16-Alyref/Gabpb1 pathway. To note, the purpose of our study is not to find a way to improve the full-term development of cloned embryos, but to reveal molecular mechanisms of the early developmental arrest of NT embryos by identifying relevant genes. Hence, we have achieved the objective of our study and our main conclusion is valid without showing the improvement of full-term development. Furthermore, from a practical point of view, the rescue of early preimplantation arrest by our identified Klf16-Alyref/Gabpb1 pathway might help to establish ES cell lines from SCNT embryos with low developmental potentials such as interspecies SCNT embryos (Azuma et al., J Reprod Dev. 2020;66:255-263), which represent the precious way for the conservation of genetic materials from endangered species. We therefore believe that our finding is scientifically valuable without showing the increased birth rate of SCNT embryos.

August 2, 2023

RE: Life Science Alliance Manuscript #LSA-2023-02296-T

Dr. Kei Miyamoto
Kindai University
Faculty of Biology-Oriented Science and Technology
930 Nishimitani
Kinokawa-shi, Wakayama-ken 649-6493
Japan

Dear Dr. Miyamoto,

Thank you for submitting your revised manuscript entitled "Incomplete activation of Alyref and Gabpb1 leads to preimplantation arrest in cloned mouse embryos". We would be happy to publish your paper in Life Science Alliance pending final revisions necessary to meet our formatting guidelines.

- please upload your main manuscript text as an editable doc file
- please upload your main and supplementary figures as single files
- please add a Summary Blurb/Alternate Abstract to our system
- please add a Category and Keywords for your manuscript to our system
- please add the Twitter handle of your host institute/organization as well as your own or/and one of the authors in our system
- please add Author Contributions to our system
- please add your main, supplementary figure, and table legends to the main manuscript text after the references section
- LSA allows supplementary figures, but no EV Figures; please update your callouts for the Supplementary Figures in the manuscript Fig EV1A=Fig S1A; while supplementary figures use the system supplementary Fig S1

A. FINAL FILES:

B. MANUSCRIPT ORGANIZATION AND FORMATTING:

Sincerely,

August 4, 2023

RE: Life Science Alliance Manuscript #LSA-2023-02296-TR

Dr. Kei Miyamoto
Kindai University
Faculty of Biology-Oriented Science and Technology
930 Nishimitani
Kinokawa-shi, Wakayama-ken 649-6493
Japan

Dear Dr. Miyamoto,

Thank you for submitting your Research Article entitled "Incomplete activation of Alyref and Gabpb1 leads to preimplantation arrest in cloned mouse embryos". It is a pleasure to let you know that your manuscript is now accepted for publication in Life Science Alliance. Congratulations on this interesting work.

DISTRIBUTION OF MATERIALS:

Again, congratulations on a very nice paper. I hope you found the review process to be constructive and are pleased with how the manuscript was handled editorially. We look forward to future exciting submissions from your lab.

Sincerely,
